



# Surface ozone trend variability across the United States and the impact of heatwaves (1990-2023)

Kai-Lan Chang[1,2], Brian C. McDonald[2], and Owen R. Cooper[2]

[1]Cooperative Institute for Research in Environmental Sciences, University of Colorado Boulder, Boulder, CO, USA
[2]NOAA Chemical Sciences Laboratory, Boulder, CO, USA

**Correspondence:** Kai-Lan Chang (kai-lan.chang@noaa.gov)

**Abstract.** This study conducts a comprehensive trend assessment of surface ozone observations across the conterminous USA over 1990-2023. A changepoint detection algorithm is applied to evaluate seasonal trends at various percentiles. Based on the results obtained from regional-scale analysis, we found that highly consistent and robust negative trends of extreme values occurred in spring, summer and fall since the 2000s across the eastern USA. A less strong, but similar picture is found in the
western USA, while increasing winter trends are commonly observed in the Southwest and Midwest. The impact of a potential climate penalty that might offset some of the improvement in the ozone extremes is also investigated based on various heatwave metrics. By comparing threshold exceedances, we found that the exceedance probabilities during heatwaves are higher than normal conditions, but the differences have decreased over time because the effectiveness of emission controls led to a great reduction of ozone extremes for both heatwave and normal conditions. When the increasing heatwave trends are accounted
for, we find evidence that the decrease of exceedance trends have likely halted during heatwave events at 20%-40% of sites. By identifying monitoring sites with (1) reliably decreasing ozone exceedances and (2) reliably increasing co-occurrences of ozone exceedances and heatwave events, we can show that several sites in California have been impacted by the ozone climate penalty (1995-2022). These findings are limited by the availability of long-term continuous ozone records, which are sparsely distributed across the USA and typically less than 30 years in length.

## 1  Introduction

Surface ozone is an air pollutant detrimental to human health (Fleming et al., 2018) and crop production (Mills et al., 2018), and an important greenhouse gas (Gaudel et al., 2018). The United States (US) Clean Air Act of 1970 requires the US Environmental Protection Agency (EPA) to set National Ambient Air Quality Standards (NAAQS) for ozone. The standards have been reviewed and updated regularly to protect the public health and welfare. The current primary standard established in 2015
and reviewed in 2020 is 70 ppbv for the fourth-highest daily maximum 8-hour ozone average (MDA8) value, averaged over three consecutive years (US EPA, 2020b).

The EPA Air Quality System (AQS) monitoring network provides extensive long-term surface observations of air pollutants and meteorology (US EPA, 2024). These data have been constantly studied to quantify the impacts of air pollution on air quality metrics, human health, and vegetation, and the data are also relevant to climate assessments (Cooper et al., 2014; Simon et al.,





2015; Fleming et al., 2018; Jaffe et al., 2018; US EPA, 2020a; Wells et al., 2021). This dataset is also an important source for evaluating climate-chemistry models and satellites (Rasmussen et al., 2012; Fiore et al., 2014; Zoogman et al., 2014). Several studies have reported the reduction of surface ozone across much of the US since the early 2000s in response to ozone precursor emission controls (Cooper et al., 2012; Simon et al., 2015; Strode et al., 2015; Seltzer et al., 2020), in which the decreases in

the eastern USA are often clearly demonstrated (Chang et al., 2017; Lin et al., 2017). Ozone across the western USA is more challenging to quantify on the regional scale due to the complex terrain, relatively sparse monitors, and impacts from wildfires and oil and gas emissions (Edwards et al., 2014; Lu et al., 2016; McDuffie et al., 2016; Parks and Abatzoglou, 2020; Francoeur et al., 2021; Buchholz et al., 2022; Langford et al., 2023; Peischl et al., 2023; Putero et al., 2023; Byrne et al., 2024; Marsavin et al., 2024; Sorooshian et al., 2024), but overall decreasing trends can be detected across the Southwest (Chang et al., 2021)

and at high-elevation rural sites (Chang et al., 2023a).

In terms of statistical modeling, tropospheric ozone variability is inherently heterogeneous in space and time. Regional trend detection of surface ozone is also complicated by irregular distribution of monitoring sites, different lengths of site records, and varying spatial coverage over time (see Fig 1 for surface ozone data availability). In the first phase of the Tropospheric Ozone Assessment Report (TOAR), a regional trend assessment for the eastern USA was conducted under the framework of the

generalized additive mixed models (GAMM), and focused on the summertime period (Apr-Sep) over 2000-2014 (Chang et al., 2017). This study presents an extensive long-term seasonal trend assessment of surface ozone observations over 1990-2023, and aims to tackle additional challenges, including (1) greater attention to the trends in ozone extremes, such as the extreme percentiles and threshold exceedances, and (2) evaluation of potential changepoints in long-term trends. Both challenges are addressed at local (individual sites) and regional scales.

While anthropogenic emission controls lead to decreasing average and extreme ozone levels, outstanding concerns remain, such as the contributions of heatwaves and wildfires to ozone production (Jing et al., 2017; Lin et al., 2017; Langford et al., 2023; WMO, 2023; Cooper et al., 2024). The IPCC Sixth Assessment Report concluded that it is virtually certain that the frequency and intensity of hot extremes and the intensity and duration of heatwaves have increased since 1950, and will continue to increase as the planet warms (Arias et al., 2021). Heatwaves provide ideal conditions for the photochemical production of

ozone due to sunny skies, low wind speeds and capped boundary layer height. Although the co-occurrence of heatwaves and ozone extremes has been identified previously (Schnell and Prather, 2017), its co-variability is difficult to explicitly quantify, particularly under rapid emission reductions. Deterioration of ozone air quality due to increasing temperatures caused by climate change is known as the climate penalty (Wu et al., 2008; Zanis et al., 2022). Projections of climate change over the 21st century suggest that a clear ozone climate penalty (annual average ozone increases of 1-3 ppbv) could emerge in high emis-

sions regions, such as northern India and eastern China, in association with large temperature increases (3°C above 1850-1900 temperatures) and on time scales of more than 40 years (Zanis et al., 2022; WMO, 2022). A final objective of our analysis is to determine if any long-term ozone monitoring sites in the USA show evidence for a climate penalty over the relatively short time scale of 1995-2022. Recognizing that detection of a climate penalty could require relatively large temperature increases, we focus on heatwave conditions during the warmest months of the year (May-September). We aim to quantify the heatwave





impact on ozone variability by comparing ozone exceedance trends between heatwave and normal conditions, under rigorous and widely adopted heatwave metrics (Perkins et al., 2012; Perkins and Alexander, 2013).

The trend detection framework of seasonal ozone percentiles and exceedances is described in Section 2. We quantify surface ozone trends at individual stations and the overall regional trends across the western/eastern USA over 1990-2023 in Section 3. After the current status of ozone trends and variability are better understood, we further discuss the potential climate penalty

for ozone extreme events in Section 4, i.e., the heatwave impact on short-term ozone variability and long-term exceedance trends. Conclusions are presented in Section 5.

## 2  Data and Methods

Because of differences in the sensitivity of ozone to precursor emissions, different monitoring sites might respond dissimilarly, in terms of the latency and magnitude of trend changes at different percentiles and seasons (Box and Tiao, 1975). Since the

presence of a trend change is suspected but its juncture is unknown, additional care needs to be taken. In this section we introduce the metrics for studying extreme ozone events, discuss the rationale for changepoint detection, describe the statistical methods for detecting trends at individual sites, and provide further considerations for deriving representative regional trends.

### 2.1  Ozone and heatwave metrics

Two types of extreme data are typically present in statistics: *block maxima* and *threshold exceedances*. The block maxima

approach, which is not based on a particular standard, originally focuses on the maximum value at each time interval (a block indicates a selected time unit, e.g., a time series of seasonal ozone maximum values (Smith, 1989)). Since the concept of extreme percentiles (e.g., 90th) is closely related to the block maxima, to avoid confusion and to represent a wider range of extreme events, we use the term *block extremes* hereafter. Threshold exceedances, also known as the peak-over-threshold, make a binary classification of data into relevant (events occur) and irrelevant (events do not occur) categories.

In this study all the ozone analyses are based on daily maximum 8-hour average (MDA8) values available from the US EPA AQS network (US EPA, 2024). The particular ozone metrics for block extremes are the 10th and 90th percentiles of MDA8 in each season (in addition to the seasonal medians); and for threshold exceedances the ozone metrics are the number of days per summertime period in which the MDA8 value exceeds the thresholds of 70, 60, 50 and 35 ppbv, mainly limited to the period between May and September (as we aim to compare the ozone exceedances between heatwave and normal conditions). Note

that the statistical characteristics are different for percentile data (continuous variable) and exceedance days (count variable), so different statistical methods are required (see Section 2.3).

To present the actual extreme ozone variability, the lower panel of Fig 1 shows the seasonal 90th percentile time series from individual stations in the western and eastern USA (based on the 100°W boundary). While ozone has generally decreased over 1990-2023 in MAM (Mar-Apr-May), JJA (Jun-Jul-Aug), SON (Sep-Oct-Nov), we can see tumultuous variability within

regions and differences between regions. For example, reductions of ozone extremes are obvious in the eastern USA, with exceedances of 70 ppbv are commonly observed before the early 2000s and infrequently after 2013. While in the western



USA, strong ozone extremes are still common in recent years, and ozone exceedances can also be observed in DJF (Dec-Jan-Feb), mainly in the snow-covered oil and gas basin of northeastern Utah (Fig S1) (Edwards et al., 2014). These features are expected to be the key factors to determine the resulting trends and uncertainty. Nevertheless, this is merely one aspect of the complexity, and full demonstrations of heterogeneous ozone variability for other percentiles (10th and 50th) are provided in Figs A1-A2. These demonstrations clearly indicate that summarizing multiple aspects of ozone variability is essential for delivering a comprehensive trend assessment.

The NOAA Physical Sciences Laboratory (PSL) and Climate Prediction Center's global unified temperature dataset (NOAA PSL, 2024) provides gridded daily temperatures at $0.5° \times 0.5°$ resolution. These gridded temperatures are interpolated to all the ozone monitor locations (interpolation details are provided in Section 2.4), and are used to determine heatwave events at each location. The heatwave metrics used in this study are adopted from well-established heatwave research literature (Perkins and Alexander, 2013; Mazdiyasni and AghaKouchak, 2015; Domeisen et al., 2023). Firstly, given daily maximum ($T_{max}$) and minimum ($T_{min}$) temperatures, the following temperature thresholds are considered:

1. TX90pct: the 90th percentile of a 15-day moving window of $T_{max}$ (centered at each calendar day); each calendar day has a different percentile value (analogous to the seasonal climatology). This is the constant temperature baseline at each grid cell or monitoring location, against which heatwave events (i.e. temperature exceedances) are defined.

2. TX95pct: same as TX90pct, but based on the 95th percentile.

3. TX35deg: a fixed threshold at 35°C for $T_{max}$.

4. TN90pct: same as TX90pct, but based on $T_{min}$.

5. TN95pct: same as TN90pct, but based on the 95th percentile.

6. TN20deg: a fixed threshold at 20°C for $T_{min}$.

An episode of heatwave event is detected if at least 3 consecutive days of temperature exceedances are found. A longer period of gridded temperature data (1990-2022) is used to better determine the temperature percentiles.

In order to investigate the heatwave impact on instantaneous ozone variability, two (short-term) statistics during heatwaves are considered for each site: (1) ozone enhancement (in units of ppbv), which is a simple MDA8 mean difference between heatwave and normal conditions, based on deseasonalized anomalies; and (2) ozone accumulation rate (in units of ppbv/day), which is designed to quantify the day-after-day rate of ozone change (if any) throughout the durations of the heatwave. The accumulation rate is our quantity of interest, because it indicates if a longer heatwave produces higher ozone concentrations. It should be noted that for each heatwave event, the initial ozone conditions are different, so it is necessary to adjust the ozone initial level between different events. This approach is also known as the random intercept model (i.e. intercepts vary for each heatwave event, but the slope or accumulation rate is invariant). Statistically, let $m_{ij}$ be the MDA8 value and $h_{ij}$ be the corresponding temporal index at $i$-th day from $j$-th heatwave event at a given site, then the model can be expressed as: $m_{ij} = (a + u_j) + bh_{ij} + residuals$, where $a$ is the overall intercept, $u_j$ is the adjustment of the initial condition for $j$-th





heatwave event, and $b$ is the daily accumulation rate during heatwave. This model is similar to trend analysis, but aligned to heatwave periods over several days (invariant to different time periods).

## 2.2 Changepoint detection applied to a large database of ozone time series database: Practical considerations

Many changepoint detection methods have been developed to evaluate optimized changepoint(s) and to model the nonlinear

trends parametrically (see reviews by Reeves et al. (2007) and Lund et al. (2023), and comparisons by Chen et al. (2011) and Shi et al. (2022)). While those methods are practical, a visual diagnosis of the time series plot is often required to validate the reliability and to avoid false positives. Therefore, an anticipated challenge for our study is that the dataset contains so many monitoring stations, that it becomes impractical to inspect the diagnostic plot for each individual time series (Fig 1). Since it has become normal to deal with a large amount of time series data for scientific assessments (Chang et al., 2021), we discuss

the practical considerations relevant to changepoint analysis of atmospheric composition data tailored to large surface ozone monitoring networks.

    The scope of our changepoint analysis focuses on the *change in long-term trends* (e.g., the presence of turnaround, or flattened trends). Addressing data shifts due to instrumentation changes is beyond the scope of our analysis. While instrumental issues might have an impact at individual sites, they should not produce regional patterns. Practical considerations for our

surface ozone analysis include:

-  Since the trend change is our primary concern, all the necessary considerations relevant to trend detection (e.g., autocorrelation and heteroscedasticity) shall be accounted for (Chang et al., 2021; Lund et al., 2023).

-  At least a few decades of observations are typically required to reliably determine long-term trends (presumably over 20 or 30 years) (Weatherhead et al., 1998; Chang et al., 2024). We aim to avoid assigning changepoint(s) to short-

term variability, because ozone is temporally variable and atmospheric circulations could induce multi-year fluctuations (Cooper et al., 2020; Fiore et al., 2022). Given that the longest record in this study is 34 years (1990-2023), to make sure that the changepoint is evaluated only through long-term changes, and not induced by multi-year fluctuations or abrupt variability near the beginning/end of the time series, we mainly consider one changepoint in our analysis and we assume the minimal segment of trends is at least 10 years, which is a useful benchmark for ensuring sufficient length of data for

trend estimation before and after the changepoint. However, the possibility of incorporating two changepoints will be evaluated, if the junctures occur separately at around 2000 and 2010.

-  Changepoint detection algorithms are typically implemented by fitting the proposed model to all possible candidates, and then the best model (assessed by the maximization of trend change or the minimization of fitted errors) indicates the optimized changepoint. Obviously, the longer the time series, the larger the candidate pool. Nevertheless, as the

specific annual/seasonal percentiles are our primary concern, we only need to consider possible candidates at the annual or seasonal scale (i.e., there is no need to identify the changepoint at an exact month).



Based on the above discussions, for each monitoring site and season, we fit the trend model to all possible changepoint candidates (between 2000-2013), and then select the best fitted result from the candidate pool, i.e., each station/season/percentile can have different changepoints (see the next section for technical details). We emphasize that for the purpose of regional changepoint detection, a strong turnaround of trends at any individual stations should not be overly generalized as representative of regional variations, rather, the justification for regional changes should be made based on consistent patterns obtained by a large cluster of stations. To evaluate the evidence for a regional trend and to evaluate the agreement between time series, the terminology provided in the guidance note on consistent treatment of uncertainties of the IPCC Fifth Assessment Report is adopted (Mastrandrea et al., 2010); details are provided in Section 3.1.

## 2.3 Percentile trend and changepoint detections for station time series

The general trend detection model consists of several components, such as seasonality, trend, and residuals (also known as time series decomposition). Since our focus is placed on specific annual/seasonal percentiles or threshold exceedances, and trend detection for the time series from all months of the year is not directly considered, seasonal adjustments are not required. Note that meteorological variables are important to attribute ozone variability at a daily or monthly scale (Chang et al., 2024), but they are not essential covariates for seasonal or annual metrics, so meteorological adjustments are also not considered. Instead, the temperature influence on ozone is investigated through the heatwave analysis in Section 4. Depending on the application, the trend component can be simply estimated by a linear form (i.e. a constant rate of change per unit), approximated by a combination of multiple linear segments (connected by changepoint(s)), or described by a complex nonlinear form (i.e. to reflect small-scale fluctuations). Even though nonlinear techniques can reveal the unique features for individual time series, their complexities make it difficult to summarize the results with a few representative trend values, especially as we need to deal with thousands of monitoring stations. Therefore, complex nonlinear trends are not adopted and we use piecewise trends to represent the major change of long-term trends (if any). Different statistical methods are used to detect trends for percentiles and exceedance days, described as follows:

- Quantile regression (Koenker, 2005; Koenker et al., 2017) is applied to derive trends in percentiles. Given a time series of observations $y_t$, where $t$ is the annual or seasonal index, and $t_c$ is the index for the changepoint candidate, then the statistical model can be expressed as:

$$y_t = \begin{cases} \beta_0 + \beta_1 t + N_t, & \text{if a linear trend over the whole period is considered,} \\ \beta_0 + \beta_1 t + \beta_2 \max(t - t_c, 0) + N_t, & \text{if a piecewise trend is considered,} \end{cases} \quad (1)$$

where $\beta_0$ is the intercept, $\beta_1$ is the trend since the beginning of the record, and $\beta_2$ is the adjustment after the changepoint $t_c$ occurred (i.e. the magnitude of the trend change). Residual term $N_t$ represents the remaining variability, which is often autocorrelated and possibly heteroscedastic. If we rewrite the residual component in Equation (1) as $N_t = y_t - \beta_0 - \beta_1 t + \beta_2 \max(t - t_c, 0)$ for the changepoint analysis, then the quantile regression finds the trend estimates through





the minimization of the residuals by the $L1$ optimization:

$$\sum_{t:N_t \geq 0}^{T} q |N_t| + \sum_{t:N_t < 0}^{T} (1-q) |N_t|, \qquad 0 < q < 1. \tag{2}$$

Changes in different percentiles can be investigated by adjusting the quantity $q$ (e.g. 0.5 represents the median and 0.9 represents the 90th percentile). This equation is a generalized form and typically more relevant for daily observations, rather than monthly aggregated data. For instance, the 90th percentile of daily observations is more representative of the relevant extreme than the 90th percentile of monthly means. However, since the focus is already placed on specific percentile time series, Equation (2) can be simplified by using a fixed $q = 0.5$, which is also known as least absolute deviations or median regression.

An empirical approach to understand the difference between conventional multiple linear regression (MLR, i.e., a mean-based method) and median regression is through the arguments between RMSE (root mean square error) and MAE (mean absolute error) (Willmott and Matsuura, 2005; Chai and Draxler, 2014; Hodson, 2022). When comparing fitted residuals among all the techniques under the same trend model, MLR is designed to produce the lowest RMSE and median regression is designed to produce the lowest MAE. Neither method can have both lowest RMSE and MAE, and both methods have their advantages and disadvantages. Nevertheless, in terms of trend detection, median regression is less sensitive to outliers than MLR, and thus is expected to provide a more robust trend estimation.

For each station/season/percentile, the trend model is fitted to all changepoint candidates ($t_c = 2000, \cdots, 2013$). We then select the one with the lowest $p$-value for the trend adjustment term $\beta_2$ as our final model (or equivalently, the greatest magnitude in signal-to-noise ratio or SNR, defined as the trend value divided by its uncertainty).

– Logistic regression is applied to derive trends in threshold exceedances. The Poisson or negative binomial regression can also be used to model count time series (Chang et al., 2017), however, the scope of our analysis includes comparisons of exceedances between heatwave and normal conditions. Since the numbers of heatwave and normal days vary in different years, their exceedance days are not directly comparable. In order to establish a common baseline, their exceedances are compared based on the (conditional) probability of an event that has occurred or not occurred. For May-Sep in each year, the following probabilities are defined and calculated:

$$P_E = P(\text{exceedance}) = \frac{\text{number of exceedance days}}{\text{number of daily observations}}, \quad P_H = P(\text{heatwave}) = \frac{\text{number of heatwave days}}{\text{number of daily observations}}, \tag{3}$$

$$P_{E \cap H} = P(\text{exceedance} \cap \text{heatwave}) = \frac{\text{number of exceedance and heatwave co-occurrence days}}{\text{number of daily observations}},$$

then we can also calculate the following conditional probabilities:

$$P_{E|H} = P(\text{exceedance}|\text{heatwave}) = \frac{P_{E \cap H}}{P_H},$$

$$P_{E|\bar{H}} = P(\text{exceedance}|\text{normal}) = \frac{P_E - P_{E \cap H}}{1 - P_H},$$





These probabilities can be interpreted as the likelihood of an ozone exceedance that occurred during heatwave or normal conditions, respectively. We can thus investigate the ozone climate penalty associated with heatwaves by comparing trends between $P_{E|H}$ and $P_{E|\bar{H}}$. The statistical model for logistic regression can be expressed as:

$$\ln\left(\frac{P(t)}{1-P(t)}\right) = \theta_0 + \theta_1 t,$$

where $t$ is a temporal index, and $P(t)$ can be a time series of the conditional or unconditional probabilities discussed above. Note that the term $\theta_1$ can not directly be interpreted as the trend value for exceedance probability, so the average marginal effect is used to represent the slope (Kleiber and Zeileis, 2008).

It is important to point out that exceedance days or probabilities are more sensitive to incomplete data, because exceedances can only be underestimated if missing values are present, while the percentiles may sometimes hold to a reasonable approximation under non-severe incompleteness. Therefore, more detailed changepoint quantifications are given to the percentiles (Section 3), and we only consider linear trends for the exceedances in the heatwave analysis (Section 4).

Presently, the standard libraries in Python and R for the implementation of quantile/median regression are designed for IID (independent and identically distributed) or heteroscedastic cases, but autocorrelation is not explicitly considered. Although the Prais-Winsten and Cochrane-Orcutt procedures (or prewhitening) have been applied to median regression (Dielman, 2005), it can severely distort the data structure in some cases (Razavi and Vogel, 2018). Therefore the moving block bootstrap approach is adopted for all trend estimates (including logistic regression) in this study to account for autocorrelation and heteroscedasticity (Fitzenberger, 1998; Lahiri, 2003). The implementation code can be found in Chang et al. (2023b).

## 2.4 Percentile changepoint detection for the overall regional trends

Spatial heterogeneity, resulting from inconsistent temporal trends and variability at different locations, must be accounted for when studying regional trends, which can be dealt with through mapping irregularly distributed measurement locations onto regular grids (see Chang et al. (2021) for detailed discussions). In this study the geostatistical modeling is implemented via the framework of the generalized additive models (GAM, Wood (2006)). Geostatistical modeling relies on certain flexible spatial correlation structures, which are fitted through the observations and can then be used to interpolate/predict values at unobserved locations. Such spatial interpolations might not always follow a Gaussian process, so validations are needed, especially for the extreme percentiles.

The class of generalized extreme value (GEV, Jenkinson (1955)) distributions was specifically proposed to model the block maxima (Smith, 1989). An incorporation of the GEV distributions into spatial modeling has been extensively implemented (Yee, 2015; Wood and Fasiolo, 2017; Youngman, 2020). It should be noted that the GEV distributions are only applied to block maxima or specific block extremes (under some modifications and additional assumptions, e.g., the annual 4th highest ozone (Berrocal et al., 2014)), and the rest of the data (i.e., other percentiles) are ignored. Therefore, an alternative approach is to derive the percentile map directly from the full data set via quantile GAM (Fasiolo et al., 2020), which is not appropriate for modeling the maxima or minima, but is applicable to the extreme percentiles. Standard GAM is an extension of multiple linear





regression, likewise, quantile GAM is an extension of quantile regression. Even though quantile GAM offers greater flexibility in terms of a wide range of percentiles, it suffers from far more intensive computational burdens than standard GAM. The above approaches (Gaussian and GEV links for standard GAM, and quantile GAM) will be compared and discussed further in Section 3.2. Regional trends are then determined based on the regional interpolations of all different seasons and percentiles.

## 3 Results: trends in ozone percentiles

In this section we carry out a changepoint analysis of long-term trends in ozone percentiles across the conterminous USA over 1990-2023. Analysis is conducted by (1) evaluating trends at individual sites, and (2) quantifying regional trends across the West and East.

### 3.1 Trends at individual stations

To reliably evaluate the changepoint of long-term trends, we select the sites with the longest time series of ozone observations
in this particular analysis, i.e., beginning no later than 1992 and extending at least to 2018 (present-day). The method is applied to different seasonal percentiles (MAM, JJA, SON, and DJF). A total of 468 sites were selected, but the seasonal percentiles are calculated only if seasonal data availability is greater than 60%. For example, most long-term sites in Oregon/Washington only operate between May-Sep, so these sites do not have sufficient data to estimate trends in MAM, SON, and DJF.

For each selected site, (1) the optimized changepoint (identified by the greatest SNR for the trend adjustment term), (2)
the trend before the changepoint ("prior-trend"), and (3) the trend after the changepoint ("posterior-trend"), are evaluated. Trend results are categorized by different scales, based on the thresholds of $p$-value at 0.01, 0.05, and 0.33 (corresponding to SNR values of $\pm 3$, $\pm 2$, and $\pm 1$, respectively). The overall confidence levels discussed below are assessed by the evidence and agreement of trends observed across the conterminous USA (see Appendix A for further discussions), instead of by the significance at individual stations (Mastrandrea et al., 2010), patterns recognized in the East/West or smaller-scale regions will
be pointed out specifically. We use the results for the seasonal 90th percentile to highlight the effectiveness of our approach (Fig 2). The complete results for the seasonal 50th and 10th percentiles are provided in Figs A3-A4, the percentages of posterior-trends by different reliability scales are summarized in Table A2, and the percentages of trend changes based on different scenarios (e.g. from positive to negative trends, or vice versa) are summarized in Table A3. The main findings are:

- For MAM, JJA, and SON, the plurality of sites show limited evidence of prior-trends in the seasonal 90th percentile,
except for the Northeast (discussed below). Nevertheless, with different changepoints detected since the 2000s (as the latent periods for emission controls may vary geographically (Box and Tiao, 1975)), reliable and consistent negative posterior-trends can be found in MAM, JJA (high agreement and robust evidence), and SON (medium agreement and robust evidence). However, at the 50th percentile, consistency of reliable negative posterior-trends can only be observed in MAM (medium agreement and robust evidence) and JJA (high agreement and robust evidence). In these cases, stronger
decreasing posterior-trends are found at the 90th percentile than the 50th percentile, indicating faster declines in extreme intensity. Ozone extreme percentiles not only have reliably decreasing posterior-trends in the majority of sites, but



also have very few reliably increasing posterior-trends. In MAM and JJA, less than 1% and 3% of sites show reliably ($p \leq 0.05$) positive posterior-trends at the 90th and 50th percentiles, respectively. While at the 50th percentile in SON, and at the 10th percentile in MAM/JJA/SON, the plurality of sites show limited evidence of posterior-trends.

- For JJA, notably, although the changepoints at many sites occur around the early 2000s in the eastern USA, a distinctive pattern is found across the Northeast: the optimized changepoints are found around 2013 and reliable negative trends are detected over 1990-~2013, while in the recent decade (~2013-2023) the trends are flat and do not show further decreases. This pattern can be attributed to strong ozone enhancements across the Northeast/Midwest in 2012, due to persistent high temperatures (Shen et al., 2016; Jing et al., 2017). Alternatively, Jiang et al. (2018, 2022) showed that the pace of NOx emissions reductions have slowed since 2010, which may lead to flattened ozone trends, as shown in the Eastern USA (Fig 1). See next section for further discussions.

- For DJF, many stations have no measurements in wintertime, so a less dense geographical coverage is present. Medium agreement and robust evidence of increasing posterior-trends can be found at the 10th percentile (41.4%). A substantial number of sites show reliable positive posterior-trends at the 90th (24.4%) and 50th (34.6%) percentiles, albeit the majority of sites show no trends ($\geq$40%). Increasing posterior-trends are mainly observed in the Northeast and the Southwest.

## 3.2 Regional trends

While the changepoint detection at individual stations provides a great deal of details regarding pattern recognition of large-scale trend changes, it is not uncommon to observe a mixture of positive and negative trends at certain sub-regions (e.g., mixed trends are commonly observed at the 10th percentile), which have become an obstacle to drawing generalized conclusions. Therefore, this section aims to tackle this issue by explicitly accounting for spatial heterogeneity and quantifying the overall regional trends across the eastern and western USA. Note that the trend results in Section 3.1 are based on selected stations where the longest records are available, but in this section all the station information (including sites with a shorter or an interrupted record in Fig 1; the same data availability criterion is applied) is used to perform geostatistical modeling and regional trend analysis.

The technical evaluations of different geostatistical modeling approaches are provided in Section S1. In summary, we find that a fast and robust estimate can be achieved by the Gaussian process approach, which is applied in the following analysis. The mapping procedure is carried out for different seasons and percentiles over 1990-2023, and the resulting regionally aggregated time series are shown in Fig 3 (trend values are provided in Table 1). The overall conclusion is that positive trends were observed in MAM, JJA and SON since 1990, but with varying turnaround points since the 2000s, these trends are found to be strongly and reliably decreasing (as expected from Figs 2 and A3-A4); while flat or weak DJF trends are observed in recent years. Specifically,

- This regional analysis provides effective quantifications of the overall trends. For example, a visual recognition of regional patterns in the western USA is challenging, because the trends are less consistent and spatial coverage is relatively





sparse compared to the eastern USA. Nevertheless, our regional analysis shows that, albeit with weaker magnitudes in the western USA, seasonal posterior-trends at different percentiles generally follow the same conclusions between the eastern and western USA.

– In MAM, JJA and SON, the 90th percentile has strong and reliable negative posterior-trends, except for the western USA in SON. The results also show that the lower the percentiles, the weaker the magnitude of the negative trends. Controls on ozone precursor emissions in the USA have been designed to reduce extreme ozone levels, and our observation that the strongest negative trends occur at the 90th percentile is consistent with those efforts.

– The strongest negative seasonal trend in this regional assessment was found in the eastern USA, at the JJA 90th percentile (-8.1 [±3.2] ppbv/decade, $p \leq 0.01$), followed by the SON 90th percentile (-6.0 [±2.8] ppbv/decade, $p \leq 0.01$), the JJA 50th percentile (-5.3 [±1.8] ppbv/decade, $p \leq 0.01$), and the MAM 90th percentile (-4.5 [±0.9] ppbv/decade, $p \leq 0.01$).

– As discussed in Section 2, we mainly consider one unknown changepoint for long-term trends over the 34-year period. However, previous studies found that US NOx emissions did not decline until the late 1990s (McDonald et al., 2012), and the pace has likely decelerated since 2010 (Jiang et al., 2018, 2022). Therefore, it is possible to consider an additional peripheral changepoint if the junctures occur separately at around 2000 and after 2010, as each trend segment satisfies a benchmark of at least 10-years in length (i.e. these changes are not likely due to short-term variability). We identify two cases that fit this scenario, which are the 90th and 50th percentiles in JJA in the eastern USA (the other cases are so variable that the second changepoint is generally unnecessary or premature). We fit the trends with two changepoints for these two time series (dotted lines in Fig 3), and find the second changepoint aligned with the flattened ozone after 2013. These two changepoints appear to explain two distinct site patterns of JJA trends in the Eastern USA in Fig 2: the plurality of sites capture the emission reductions around 2000 (the primary change), and some sites coincide with strong ozone enhancements across the Northeast/Midwest in 2012 and flattened ozone trends after 2013 (the secondary change). Since Fig 2 already identified which sites have a changepoint around 2013, it seems unnecessary to conduct a duplicate analysis by considering two-changepoint modeling for individual sites, given that this particular analysis is reasonable partly due to the regional assessment providing a more clear signal.

– Strong heterogeneity in seasonal trends across the eastern USA substantially changes the ozone seasonality. We compare the regional monthly percentiles over different periods in Fig 4, based on the same geostatistical modeling approach, and based on individual year and multi-year averages. We can clearly see that the seasonal peaks at the 90th and 50th percentiles shifted from summer in 1990-1999 to spring in 2013-2023 (this period is chosen because JJA ozone tends to be steady over 2013-2022), along with increased ozone in DJF and decreased ozone in other seasons. In the western USA the shape of the seasonal cycles remains similar, although the 90th percentile clearly decreased from 2000-2012 to 2013-2023 during MAM/JJA. A modeling study by Clifton et al. (2014) showed that under continued emission controls, surface ozone seasonal peaks are expected to shift to Feb-Mar by the end of the 21st century over the Northeast, the current peak in April suggests that observed seasonal cycle is moving in the direction predicted by Clifton et al. (2014).





– For the 6-month warm season (Apr-Sep), a previous regional trend study found an overall mean MDA8 trend of -0.43 [$\pm$0.28] ppbv/year ($p \leq$0.01) in eastern North America over 2000-2014 (Chang et al., 2017). Albeit with a different domain, a trend value of -0.39 [$\pm$0.16] ppbv/year ($p \leq$0.01) is found in eastern USA over 2000-2023. The corresponding trend in the western USA is -0.16 [$\pm$0.10] ppbv/year ($p \leq$0.01).

This analysis summarizes the current status of consistent decreases in ozone extremes at the regional scale. While the
reductions are apparent and profound, some abnormally high ozone levels can be observed in recent years (e.g. 2018/2021 in the West and 2023 in the East), also indicated by the spatially interpolated maps of the 90th percentile in JJA (Fig S3), which can be associated with extensive wildfires seasons and/or heatwaves (Bartusek et al., 2022; Langford et al., 2023; Rickly et al., 2023; Cooper et al., 2024). As wildfires and heatwaves are expected to continue to increase, they will likely impact surface ozone levels across the USA in the near future (see next section for further discussions).

**4    Results: Potential ozone climate penalty due to heatwaves**

In the previous section we concluded that strong reductions in regional surface ozone extremes occurred across much of the USA over the past two decades. In this section we aim to investigate potential short-term and long-term impacts of heatwave events on ozone exceedances. Since we expect to classify a majority of data into normal days and a minority of data into heatwave days, a comparison based on the extreme percentiles might not be feasible due to fewer samples during heatwave
events, therefore the probability of threshold exceedances is used in the analysis. We first evaluate the distribution and trends in ozone exceedances at different thresholds based on all daily MDA8 ozone values during May-Sep, then investigate the heatwave trends, and finally compare ozone exceedances between heatwave and normal days.

This analysis is focused on the period 1995-2022, because (1) this period is a balance between an inclusion of more sites (25+ years in Fig 1) and a sufficiently long period (nearly 30 years); (2) ozone exceedances have consistently decreased in
this period (Section 4.1), so we can investigate if heatwave events counteract the progress of emission controls. The possible deceleration of emission trends should not play an important role here, as its influence is expected to be similar between heatwave and normal conditions; and (3) 2023 is an anomalous year with many ozone exceedances across the Upper Midwest and the Northeast due to large forest fire plumes transported from Canada (Cooper et al., 2024). These large amounts of exceedances are not observed in the previous decade (2013-2022). Therefore, the 2023 data are excluded to avoid skewing our
trend estimates.

**4.1    Distribution and trends in ozone exceedances**

To give an overview of the current distribution of ozone exceedances, Fig 5 shows the average exceedance days and probabilities ($P_E$ in Equation (3)) in May-Sep across the conterminous USA, based on the thresholds of 70, 60, 50 and 35 ppbv, and the periods 1995-1999 and 2018-2022 (multi-year averages are taken to ensure representativeness). Exceedance days above 70
ppbv in the eastern USA in the early period have greatly diminished in the recent period. Southern California is the only cluster where greater than 30 ozone exccedance days per year are found at multiple sites in the present-day period (2018-2022). The





reductions of ozone above 60 and 50 ppbv in the eastern USA are also obvious. Changes in the pattern of exceedances above 35 ppbv are less profound, but a general reduction can be observed in the eastern USA. Broadly speaking, for the threshold of 50 or 35 ppbv, the number of exceedances across the Southwest has remained relatively constant (except along the California coast).

We further show the trends in ozone exceedances over 1995-2022 in Fig 6, based on the probability measure (a change rate of 1 %/year corresponds to 1.53 days/year, given that the total number of days is constant in May-Sep). As expected, the widespread distribution of reliably negative trends at the thresholds of 70, 60 and 50 ppbv reinforces the fact that ozone extremes are decreasing across the USA (except for a few sites in the Southwest). In these panels (mainly for 70 ppbv and some for 60 ppbv) we use right angle arrows to indicate when very few average exceedance days ($< 3$ days) occur in the present-day, and thus the normal arrowheads identify sites with room to improve. It should be noted that the exceedance trends eventually become flat or less negative once very low exceedances are reached for several years. Therefore, even if the ozone concentrations in the extreme percentiles are continuously decreasing, the exceedance trends cannot reflect such decreases once the concentrations are below the threshold. As a result, the negative trends are generally stronger at 60 ppbv than at 70 ppbv.

In summary (Table S2), high agreement and robust evidence of decreasing exceedances at different thresholds is found across much of the USA: greater than 78% of sites show reliable decreases ($p \leq 0.05$) in threshold exceedances based on 70, 60, and 50 ppbv, less than 1% of sites have reliably positive trends at the thresholds of 70 and 60 ppbv, and 3.1% of sites have reliably positive trends at the threshold of 50 ppbv. As for the 35 ppbv threshold exceedances, 6.3% of sites have reliably positive trends and 54.8% of sites have reliably negative trends.

## 4.2 Heatwave metrics and short-term impact on ozone

To gain a quick insight into the temperature variability, we assess trends in $T_{max}$ and $T_{min}$ (Fig S5), and trends in heatwave frequency (the number of heatwave days per May-Sep) based on different heatwave metrics (Fig S6). Despite certain clusters of positive trends being detected, inconsistent spatial patterns of trends are observed between $T_{max}/T_{min}$ and different heatwave metrics. It is thus important to acknowledge that temperature trends are also heterogeneous, so their assessed impact on ozone should not be based solely on a single heatwave metric. Nevertheless, by using GAM methodology and interpolating gridded daily temperature data onto all ozone monitoring sites, we find substantially stronger correlations between MDA8 and $T_{max}$ than MDA8 and $T_{min}$ (Fig S7). Therefore, we only consider heatwave metrics based on $T_{max}$ hereafter.

It should be noted that Fig 6 represents all available long-term sites over 1995-2022, however, the higher the temperature thresholds, the fewer the number of sites that qualify (e.g. extreme temperatures are present, but have not occurred consecutively). To ensure that a site has enough data for accurate trend detection, we only show the results when heatwaves occurred in at least 10 years between 1995-2022. The implication is that since a large portion of sites have insufficient records that meet the TX35deg metric, the TX35deg results are merely provided for a reference, and the main conclusions are based on the TX90pct/TX95pct metrics.



The results for short-term ozone statistics based on different heatwave metrics are shown in Fig 7. Positive ozone enhancements can be observed consistently across most locations, with an overall average enhancement (standard deviation or SD) of 5.9 (SD = 2.9), 6.3 (SD = 3.2), and 9.9 (SD = 6.2, subject to fewer sites) ppbv for the TX90pct, TX95pct, and TX35deg metrics, respectively. Although a large portion of sites show no evidence ($p > 0.33$) of ozone accumulations (Table S1), we find that around 30% of sites in the western USA have reliable positive accumulations ($p \leq 0.05$), based on the TX90pct and TX95pct metrics. For the eastern USA, reliable positive accumulations only occurred at around 10% of sites, and are mainly observed in the Southeast. In summary, ozone deteriorations during heatwave events can be generally anticipated across the western and South-Central USA.

## 4.3 Ozone exceedances in heatwave and normal conditions

In this section we investigate if the long-term trends of ozone exceedances exhibit any different patterns when the analysis is constrained to heatwave observations (by partitioning the time series into two mutually exclusive sets: heatwave and normal days). The aims of this analysis are to (1) quantify the differences of exceedance probabilities (EP) between heatwave and normal conditions, and (2) evaluate if the effectiveness of emissions controls can be hampered during heatwave events.

Similar to the unconditional probability ($P_E$) in Fig 5, we show the EP conditioned on heatwave and normal days based on the TX90pct metric in Fig 8 (denoted by $P_{E|H}$ and $P_{E|\bar{H}}$, respectively). Similar patterns between $P_E$ and $P_{E|\bar{H}}$ can be observed for different thresholds and periods. As expected, $P_{E|H}$ is generally greater than $P_{E|\bar{H}}$ at different thresholds, and also clearly decreased between 1995-1999 and 2018-2022. The above results also hold for the TX95pct and TX35deg metrics (Figs S8-S9). To give an overall comparison and summary of Figs 6 and 8, we calculate the annual May-Sep averages of $P_E$, $P_{E|H}$, and $P_{E|\bar{H}}$ across all long-term sites (by a simple average, no weighting is applied), and estimate the trends in Fig 9. In terms of probability theory, if $P_{E|H} > P_E$, it implies that ozone exceedances and heatwaves are positively correlated (see Section S2). Nevertheless, the difference between $P_{E|H}$ and $P_{E|\bar{H}}$ provides a more clear indication about the magnitude of enhanced EP during heatwaves. The findings can be summarized as follows:

– For the threshold of 70 ppbv, the baseline EP ($P_{E|\bar{H}}$) is around 15% in the early period and 1-4% since 2013, with a trend of -4.9 [±1.9] %/decade. $P_{E|H}$ is substantially greater than $P_{E|\bar{H}}$ in the early period, but the differences are reduced from 18% in 1995-1999 to 4% in 2018-2022.

– For the thresholds of 60 and 50 ppbv, heatwaves increase the EP by 14% and 16% on average ($p \leq 0.01$, based on paired $t$-test), respectively, but the differences are reducing over time since $P_{E|H}$ show stronger decreasing trends than $P_{E|\bar{H}}$.

– Note that since the baseline EP for 35 ppbv (typically more than 70% of days in May-Sep) is much higher than 50 and 60 ppbv in normal conditions, naturally it restricts the room for additional enhanced EP during heatwaves (bounded by 1). We find that heatwaves increase the EP by 9% on average ($p \leq 0.01$) for the threshold of 35 ppbv, and do not find evidence that their differences have changed over time.





We show the distribution of the enhanced EP (annual estimates of $P_{E|H} - P_{E|\bar{H}}$ over 1995-2022) under different heatwave metrics in the top row of Fig 10. Overall, similar conclusions can be drawn for the TX95pct and TX35deg metrics (also see Figs S10-S11).

The major limitation of the above analysis is that heatwave trends and variability are not taken into account. This technical challenge is because $P_{E|H}$ is undefined when $P_H = 0$, implying that $P_{E|H}$ can only be calculated when a heatwave has already occurred. Therefore, the above analysis does not reflect the reality that heatwave events may be fewer in the early period and more frequent in the present-day. In order to account for the heatwave variability, we estimate exceedance trends at individual sites based on the logistic regression, which also allows $P_H = 0$ in certain years. The hypothesis is that as heatwaves have become more frequent they are likely to contribute to additional ozone exceedances, slowing the progress of decreasing the frequency of ozone exceedances. By comparing the exceedance trends in normal and heatwave conditions, we are able to quantify how many reliable decreasing exceedance trends are sustainable under heatwave conditions. The results are summarized in the bottom row of Fig 10: we find that reliable decreasing exceedance trends cannot be maintained for around 20%-30% of sites at the thresholds of 70, 60, and 50 ppbv, and for around 40% of sites at the threshold of 35 ppbv. A more detailed view of exceedance trends in normal and heatwave conditions are provided in Figs S12-S14 (and discussed in Section S3). In summary, these findings imply that heatwave events not only increase the EP, but also likely counteract the decreasing exceedances by its increasing frequency.

Although the TX90pct metric is previously recommended to study heatwaves (Perkins and Alexander, 2013), this metric might not necessarily be ideal for correlating heatwave with ozone. The advantages of the TX90pct are that its threshold is tailored to different environments and it thus ensures sufficient heatwave observations. However, the 90th percentile might not represent the extreme temperature condition, especially for mid- and high latitudes. Our findings show that the higher the temperature threshold, the fewer the reliably negative exceedance trends, and also the less data to meet the heatwave condition. Therefore the results are subject to the limitation of fewer samples being used for trend detection in heatwave conditions. As a longer time period makes estimations of higher temperature thresholds more reliable, future studies are recommended to use multiple heatwave thresholds to quantify heatwave impact on ozone.

For our final analysis we identify the monitoring sites that have experienced a clear ozone climate penalty over the relatively short time period of 1995-2022. We derived trends in co-occurrences ($P_{E\cap H}$) based on different ozone thresholds and heatwave metrics (Fig S15), and found that positive $P_{E\cap H}$ trends ($p \leq 0.05$) can be commonly observed in the West Coast. However, we do not find similar evidence that the correlations between ozone exceedances and heatwaves have increased (Fig S16). The reason behind this is that the correlations are jointly determined by $P_{E\cap H}$, $P_E$ and $P_H$, thus trends in correlations can be counterbalanced by stronger decreasing $P_E$ and weaker increasing $P_H$ and $P_{E\cap H}$. Therefore, positive trends in co-occurrences do not necessarily lead to increasing correlations between ozone exceedances and heatwaves. To identify the locations with a clear climate penalty, we searched for the monitoring sites which show decreasing ozone exceedances ($p \leq 0.05$), but have increasing co-occurrences of exceedances and heatwaves ($p \leq 0.05$) over the same period (Fig 11). In this map we used 70 or 60 ppbv as the ozone threshold, and all three heatwave metrics were considered, but we only showed the results when at least two out three heatwave metrics revealed a likely climate penalty. We identified 19 sites that have experienced a climate penalty





since 1995. Co-occurrence time series of 70 ppbv threshold exceedances and heatwaves are shown at four of these locations (Fig S17). All of these sites are in California and are associated with daily maximum temperature increases of 1-3°C over the three decades from 1990 to 2022 (May-Sep only) (Fig 11). Many states in the western USA have regions with similar or greater temperature increases but we did not find evidence for a climate penalty. However, our study is limited by the fact that ozone monitoring is sparse in many of these regions and therefore the climate penalty cannot be thoroughly evaluated.

## 5    Conclusions

By quantifying trends in MDA8 ozone extreme percentiles and threshold exceedances, this study shows robust and consistent evidence of decreasing ozone extreme events across much of the USA, previously attributed to strict controls of anthropogenic emissions (Cooper et al., 2012; Simon et al., 2015; Strode et al., 2015; Lin et al., 2017; Jin et al., 2020; Seltzer et al., 2020). The evidence is supported by (1) reliably negative seasonal trends at the 90th percentile in MAM, JJA, and SON from different
changepoints since 2000 (-4∼-8 ppbv/decade in the East and -1∼-4 ppbv/decade in the West, dependent on seasons); and (2) threshold exceedances of 70, 60, 50 and 35 ppbv also show strong reductions in May-Sep over 1995-2022.

Incorporating changepoint(s) in the trend detection model is also known as an intervention analysis in statistics (Box and Tiao, 1975), as it enables us to quantify the effectiveness from an intervention, e.g., anthropogenic emissions control. By focusing on the results after the changepoints, our findings can be summarized as follows:

– Based on individual long-term sites (1990-2023), we find that at least 46% of sites have reliably negative trends and at most 3% of sites have reliably positive trends at each of the following seasonal percentiles: the 90th percentile in MAM/JJA/SON, and at the 50th percentile in MAM/JJA (medium to high agreement). In contrast, reliably positive DJF trends are observed more commonly at the lower percentiles, 41%, 35% and 24% of sites at 10th, 50th and 90th percentiles, respectively, albeit subject to lower site availability than other seasons.

– Based on regional geostatistical modeling, we found the higher the percentiles, the stronger the negative regional trends in MAM/JJA/SON. Robust negative trends ($p \leq 0.01$) can be identified in MAM and JJA at the 50th/90th percentiles in both the western and eastern USA, while in SON robust negative trends can only be found at the 90th percentile in the East. Strong decreasing JJA trends in the East substantially change the ozone seasonality: the seasonal peaks shifted from summer in the 1990s to spring in the most recent decade.

In addition, in response to the shift of NOx emissions trends from a rapid decline since the late 1990s to a slowdown after 2010 (McDonald et al., 2012; Jiang et al., 2018, 2022), we explored the possibility of incorporating two changepoints for trend detection. We found that this shift fits the summer ozone variability in the Eastern USA (the 90th and 50th percentiles). The primary changepoint around 2000 corresponds to substantial ozone decreases and the secondary changepoint at 2013 corresponds to flattened ozone trends (Fig 3). This pattern is currently not identified in other seasons and percentiles, but it
warrants future modeling studies.





To deliver an observational-based assessment of heatwave impact on ozone extremes, we compare MDA8 exceedance probabilities and trends based on different heatwave metrics. In summary, the effectiveness of emission controls leads to substantial reductions of exceedances above 70, 60, and 50 ppbv. Although the exceedance probabilities are typically higher during heatwaves than normal conditions, the differences are substantially reduced in the present-day period (2018-2022). We also find that reliable decreasing exceedance trends are likely to have halted at 20%-40% of sites during heatwaves, dependent on the ozone thresholds and heatwave metrics. Therefore, heatwaves not only increase the exceedance probabilities, but also likely

counteract the effectiveness of emission controls. The limited number of sites that have experienced a clear ozone climate penalty since 1995 are all located in California, where the ozone monitoring network is dense. The sparse ozone monitoring network across the other regions of the western USA, where temperatures have increased greatly, hinders our ability to identify other locations that may have experienced a climate penalty. This climate penalty analysis is also limited because the available ozone time series are less than 30 years in length, while the chemistry-climate model projections for identifying regions most

likely to experience an ozone climate penalty are based on much longer time scales (Zanis et al., 2022). Apart from heatwaves, recent studies also show that the increasing occurrence of wildfires could be an important factor for ozone exceedances in the present-day and near future (Langford et al., 2023; WMO, 2023; Cooper et al., 2024; Seguel et al., 2024).

## Appendix A:  Further discussions and additional results for seasonal percentile changepoint analysis

In Section 3.1 our focus has been on the seasonal extreme percentile, this section aims to provide more detailed discussions on

the confidence levels for a trend assessment, and analysis for additional percentiles (Figs A1-A2). In the main text we applied geostatistical modeling techniques to summarize the regional trends (Section 3.2). An alternative approach can be used based on the evidence and agreement obtained from individual sites. Although the method and terminology are based on Mastrandrea et al. (2010), the implementation details need to be tailored to our study (Table A1):

- Evidence: The reliability of trend estimate is classified into 5 scales, including robust (positive/negative trends with

20           $p \leq 0.05$), medium (positive/negative trends with $0.05 < p \leq 0.33$), and limited (trends with $p > 0.33$) evidence.

- Agreement: Long-term sites are classified according to their trend evidence. The consistency is determined by which scale has the highest percentage of sites, and is ranked from high ($> 50\%$ of sites are classified as the same scale), medium ($33\% - 50\%$ of sites), to low ($< 33\%$ of sites).

This table is used to assign the confidence level to the trend assessment of individual long-term sites (Table A2 and Figs

2, A3-A4). Since geostatistical modeling techniques are more effective to explicitly quantify the regional trends, we do not further separate the West/East in Table A2. In addition, instead of using $p$-value to determine the reliability of the changepoint, a more scientifically meaningful approach is to categorize different scenarios of trend alterations before/after the changepoint. We consider trend alterations based on the following scenarios (Table A3):

A.  P→N: from reliable positive ($p \leq 0.05$) to reliable negative ($p \leq 0.05$) trends;



B. W→N: from weak ($p > 0.05$) to reliable negative ($p \leq 0.05$) trends;

    C. N→P: from reliable negative ($p \leq 0.05$) to reliable positive ($p \leq 0.05$) trends;

    D. W→P: from weak ($p > 0.05$) to reliable positive ($p \leq 0.05$) trends;

    E. P→W: from reliable positive ($p \leq 0.05$) to weak ($p > 0.05$) trends;

    F. N→W: from reliable negative ($p \leq 0.05$) to weak ($p > 0.05$) trends.

All the other scenarios (e.g. the same evidence scale before/after the changepoint, or transitions between weak positive and
weak negative) are considered to be no evidence of trend changes. With this approach, the results show that the plurality of
sites are classified as 1) no evidence in DJF and Scenario B in other seasons for the 90th percentile, 2) Scenario B in JJA and
no evidence in other seasons for the 50th percentile, and 3) Scenario D in DJF and no evidence in other seasons for the 10th
percentile. The above method to determine level of agreement can also be applied here (not shown).

*Code and data availability.* The EPA AQS ozone data are publicly available at https://aqs.epa.gov/aqsweb/airdata/download_files.html (US
EPA, 2024). The CPC (Climate Prediction Center) Global Unified Temperature data are provided by the NOAA PSL at https://psl.noaa.gov/
data/gridded/data.cpc.globaltemp.html (NOAA PSL, 2024). The R packages used in trend estimations include mgcv (Wood, 2006), qgam
(Fasiolo et al., 2020), and quantreg (Koenker, 2005). The implementation of moving block bootstrapping for quantile regression is provided
in Chang et al. (2023b).

*Author contributions.* KLC contributed to the conception/design and conducted the analysis. KLC and ORC drafted the paper, while BCM
helped with the revision. All authors approved the submitted and revised versions for publication.

*Competing interests.* At least one of the (co-)authors is a guest member of the editorial board of Atmospheric Chemistry and Physics for the
special issue "Tropospheric Ozone Assessment Report Phase II (TOAR-II) Community Special Issue (ACP/AMT/BG/GMD inter-journal
SI)". The authors have no other competing interests to declare.

*Acknowledgements.* KLC is supported by NOAA cooperative agreement NA22OAR4320151.





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





**Table 1.** Ozone trends [$\pm2\sigma$ and $p$-value] at the seasonal 90th, 50th, and 10th percentiles in the western and eastern USA, including the optimized changepoint (CP), prior-trends (1990-CP) and posterior-trends (CP-2023).

| Pct | Season | Western USA | | | | | Eastern USA | | | | |
|---|---|---|---|---|---|---|---|---|---|---|---|
| | | prior-trends | $p$-value | CP | posterior-trends | $p$-value | prior-trends | $p$-value | CP | posterior-trends | $p$-value |
| 90th | MAM | 1.6 [$\pm$2.1] | 0.15 | 2008 | -3.8 [$\pm$1.7] | $\leq$0.01 | 4.0 [$\pm$6.9] | 0.25 | 2001 | -4.5 [$\pm$0.9] | $\leq$0.01 |
| | JJA | 4.4 [$\pm$5.3] | 0.11 | 2001 | -3.1 [$\pm$1.8] | $\leq$0.01 | 0.9 [$\pm$9.8] | 0.85 | 2000 | -8.1 [$\pm$3.2] | $\leq$0.01 |
| | SON | 2.1 [$\pm$5.2] | 0.42 | 2003 | -1.4 [$\pm$2.2] | 0.22 | 6.9 [$\pm$7.0] | 0.06 | 2000 | -6.0 [$\pm$2.8] | $\leq$0.01 |
| | DJF | 6.1 [$\pm$2.4] | $\leq$0.01 | 2000 | -0.1 [$\pm$0.7] | 0.77 | 1.5 [$\pm$1.0] | 0.01 | 2011 | -0.3 [$\pm$2.2] | 0.79 |
| 50th | MAM | 2.3 [$\pm$1.4] | $\leq$0.01 | 2008 | -2.7 [$\pm$1.0] | $\leq$0.01 | 2.3 [$\pm$2.4] | 0.07 | 2006 | -3.0 [$\pm$1.4] | $\leq$0.01 |
| | JJA | 6.0 [$\pm$4.7] | 0.02 | 2000 | -2.1 [$\pm$1.4] | $\leq$0.01 | 2.2 [$\pm$6.2] | 0.49 | 2001 | -5.3 [$\pm$1.8] | $\leq$0.01 |
| | SON | 3.0 [$\pm$3.5] | 0.09 | 2003 | -0.5 [$\pm$1.2] | 0.42 | 4.5 [$\pm$5.4] | 0.10 | 2000 | -0.9 [$\pm$1.6] | 0.25 |
| | DJF | 8.0 [$\pm$3.7] | $\leq$0.01 | 2000 | 0.1 [$\pm$0.8] | 0.78 | 2.7 [$\pm$1.0] | $\leq$0.01 | 2011 | 0.6 [$\pm$2.1] | 0.54 |
| 10th | MAM | 3.0 [$\pm$1.4] | $\leq$0.01 | 2008 | -2.7 [$\pm$1.6] | $\leq$0.01 | 2.2 [$\pm$1.6] | $\leq$0.01 | 2007 | -2.0 [$\pm$1.5] | $\leq$0.01 |
| | JJA | 5.1 [$\pm$5.8] | 0.09 | 2001 | -0.9 [$\pm$1.2] | 0.14 | 2.1 [$\pm$2.6] | 0.13 | 2001 | -2.3 [$\pm$1.0] | $\leq$0.01 |
| | SON | 2.6 [$\pm$2.0] | $\leq$0.01 | 2005 | 0.1 [$\pm$1.2] | 0.93 | 3.1 [$\pm$3.8] | 0.11 | 2000 | 0.8 [$\pm$0.9] | 0.11 |
| | DJF | 7.2 [$\pm$4.9] | $\leq$0.01 | 2000 | 0.5 [$\pm$1.1] | 0.33 | 3.5 [$\pm$0.8] | $\leq$0.01 | 2013 | -0.1 [$\pm$3.4] | 0.97 |



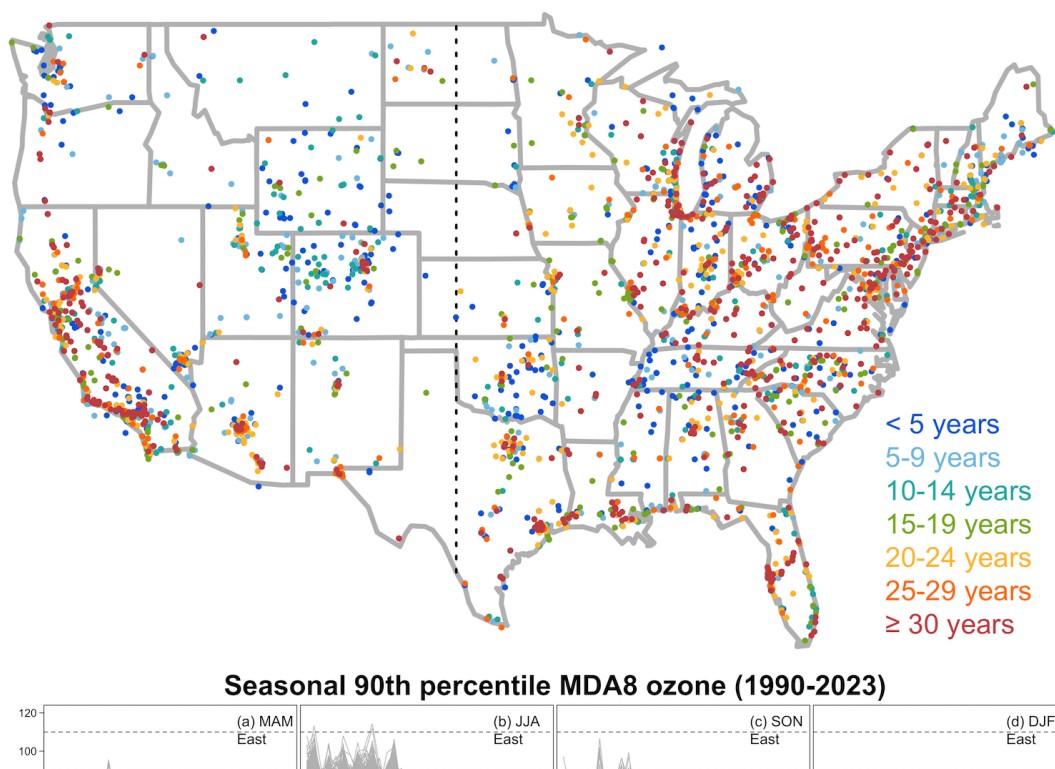

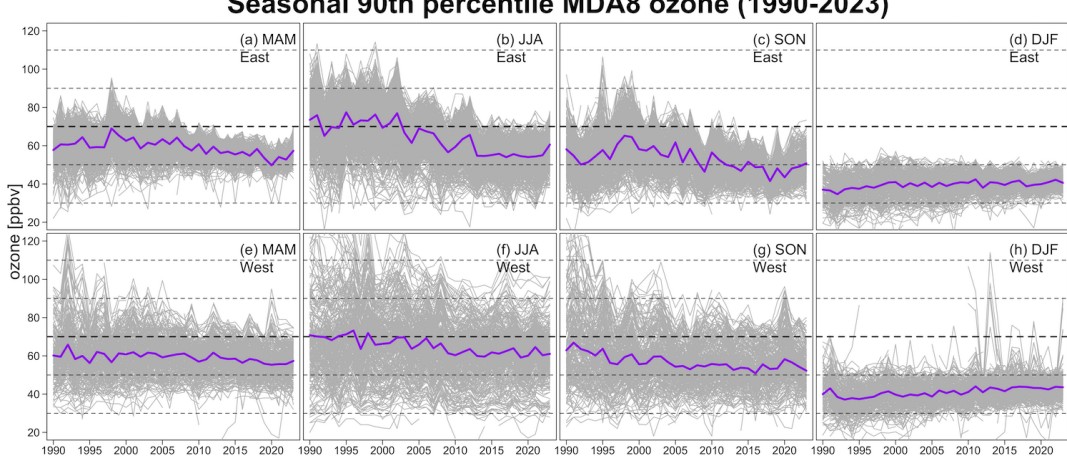

**Figure 1.** The upper panel shows data availability of the EPA AQS surface ozone observations over 1990-2023, and a total of 2540 sites are shown. The lower panel shows MDA8 time series at the seasonal 90th percentile: observations from individual stations are shown in gray, and simple averages are shown in purple.





**Figure 2.** MDA8 trends prior (first column) and posterior (second column) to the changepoints (third column) for the seasonal 90th percentile over 1990-2023. For each vector, the direction indicates the trend magnitude and the color indicates its *p*-value. The dot colors indicate the year of the changepoint.





**Figure 3.** Regional MDA8 trends in the seasonal 90th, 50th, and 10th percentiles, based on daily MDA8 values from all available sites, in the western and eastern USA (1990-2023). For the JJA 90th and JJA 50th percentiles in the eastern USA, trends based on one changepoint (solid lines) or two changepoints (dotted lines) are shown.



**Figure 4.** Regional MDA8 monthly percentiles over different years (thin lines) and periods (bold lines) in the western and eastern USA (1990-2023).





**Figure 5.** MDA8 exceedance days (top) and probabilities (bottom) based on various ozone thresholds in the early period (1995-1999) and present-day (2018-2022).



**Figure 6.** Trends [%/year] in MDA8 exceedance probabilities based on various ozone thresholds (May-Sep, 1995-2022).





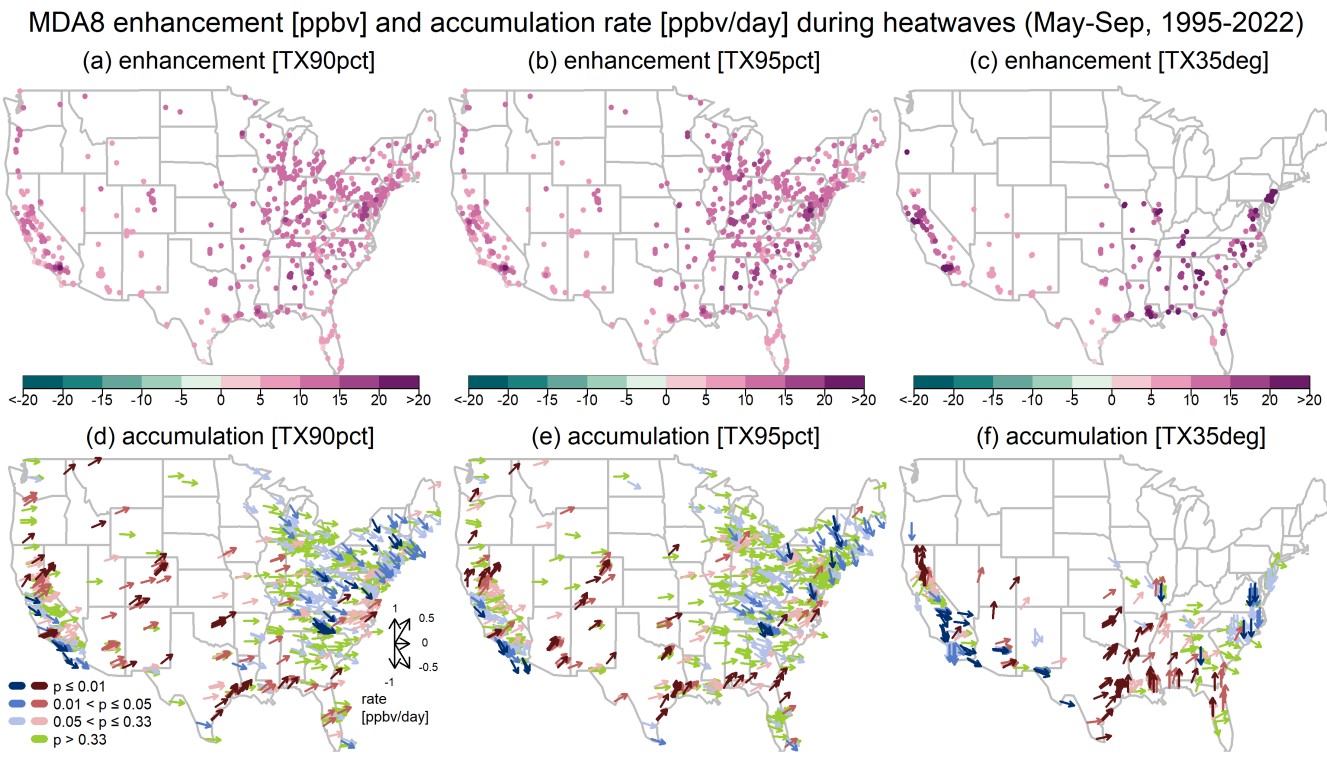

**Figure 7.** MDA8 average differences between heatwave and normal conditions (enhancement, top panel) and the rate of daily ozone changes when heatwaves events continue/extend (accumulation, bottom panel), based on various heatwave metrics (May-Sep, 1995-2022).







**Figure 8.** Conditional probabilities of MDA8 exceedances in May-Sep, based on various ozone thresholds, and average over (top) 1995-1999 and (bottom) 2018-2022.



**Figure 9.** Summary of trends in unconditional and conditional exceedance probabilities, based on various ozone thresholds and the TX90pct metric (May-Sep, 1995-2022). Squares indicate unconditional probabilities. Upward and downward triangles indicate conditional probabilities in normal and heatwave days, respectively.



**Figure 10.** Summary of (a) enhanced exceedance probabilities (each bar is made by 28 estimates of $P_{E|H} - P_{E|\bar{H}}$ over 1995-2022, averages over 1995-1999 and 2018-2022 are shown in purple and orange crosses, respectively), and (b) percentages of reliable decreasing exceedance trends ($p \leq 0.05$) in normal and heatwave conditions, based on various ozone thresholds and heatwave metrics (May-Sep, 1995-2022).





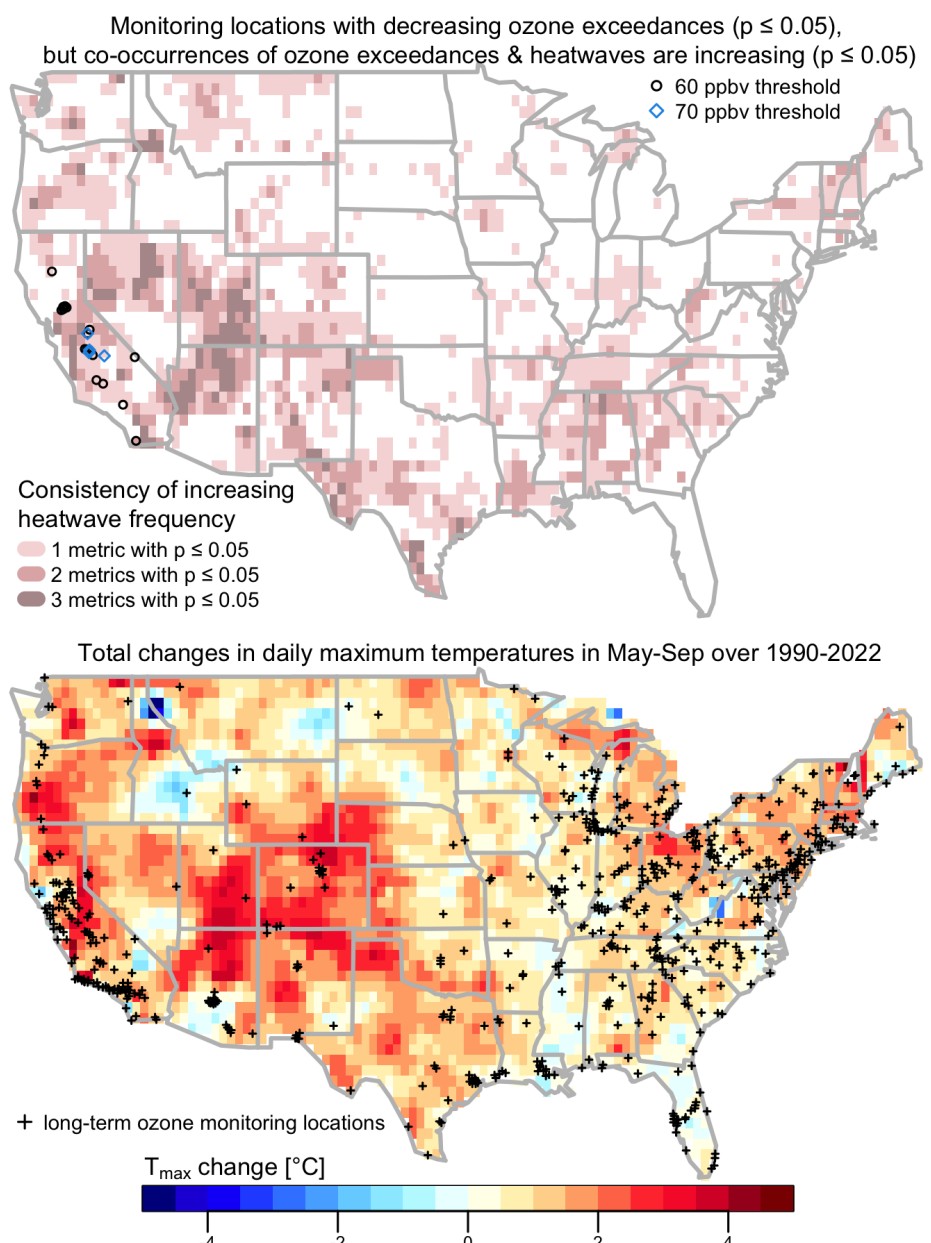

**Figure 11.** The upper panel shows ozone monitoring locations with a clear indication of climate penalty: selected sites show decreasing ozone exceedances above 70 and/or 60 ppbv ($p \leq 0.05$), but increasing trends in co-occurrences of ozone exceedances and heatwaves are detected ($p \leq 0.05$) based on at least two out of three heatwave metrics. The background indicates increasing heatwave trends ($p \leq 0.05$) detected by one (pink), two (red) or three (brown) heatwave metrics. The lower panel shows estimated increases of daily maximum temperatures in May-Sep over 1990-2022. Black crosses indicate long-term monitoring locations used in the heatwave analysis.

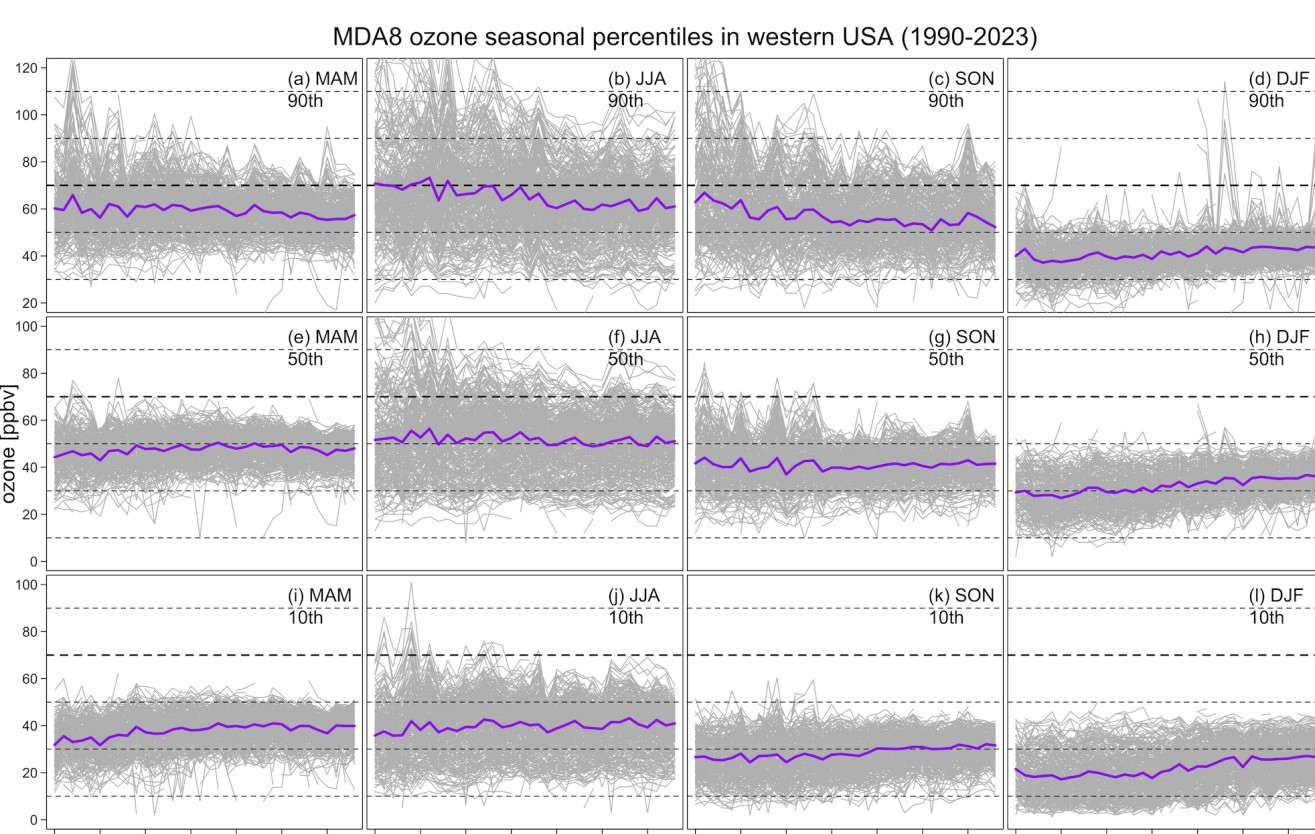

**Figure A1.** Ozone time series at the seasonal 90th, 50th, and 10th percentiles in the western USA: observations from individual stations are shown in gray, and simple averages are shown in purple.

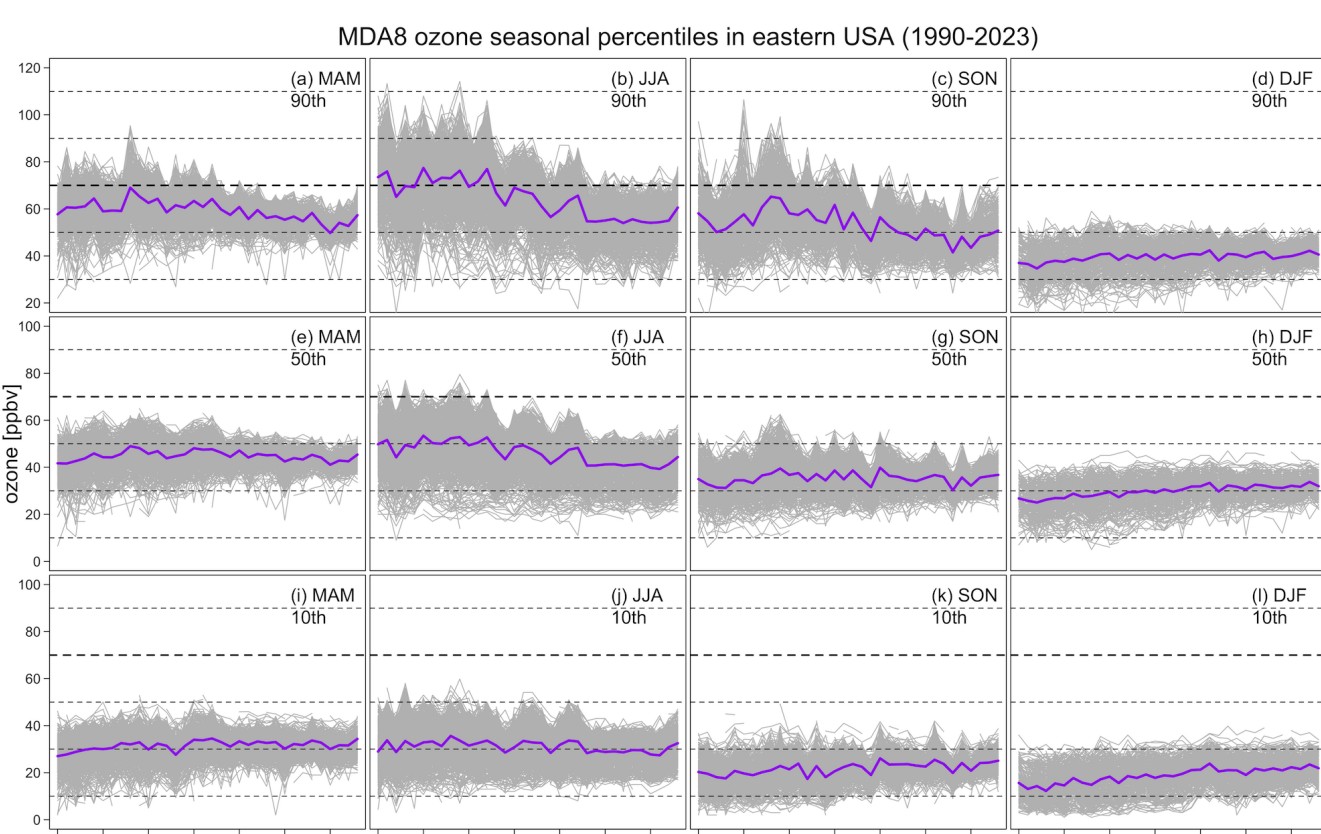

**Figure A2.** Ozone time series at the seasonal 90th, 50th, and 10th percentiles in the eastern USA: observations from individual stations are shown in gray, and simple averages are shown in purple.



**Figure A3.** Same as Fig 2, but for the seasonal 50th percentile.



**Figure A4.** Same as Fig 2, but for the seasonal 10th percentile.



**Table A1.** Criteria to determine agreement and evidence. The level of agreement is determined by which trend evidence scale has the highest percentage of sites.

| High agreement ($\geq 50\%$ site consistency) Limited evidence ($p >0.33$) | High agreement ($\geq 50\%$ site consistency) Medium evidence ($0.33\leq p <0.05$) | High agreement ($\geq 50\%$ site consistency) Robust evidence ($p \leq 0.05$) |
|---|---|---|
| Medium agreement ($[33\%, 50\%)$ site consistency) Limited evidence ($p >0.33$) | Medium agreement ($[33\%, 50\%)$ site consistency) Medium evidence ($0.33\leq p <0.05$) | Medium agreement ($[33\%, 50\%)$ site consistency) Robust evidence ($p \leq 0.05$) |
| Low agreement ($< 33\%$ site consistency) Limited evidence ($p >0.33$) | Low agreement ($< 33\%$ site consistency) Medium evidence ($0.33\leq p <0.05$) | Low agreement ($< 33\%$ site consistency) Robust evidence ($p \leq 0.05$) |

**Table A2.** Site percentages of posterior-trends by different reliability scales in Figs 2, A3 and A4 (trends with $p \leq 0.01$ are merged into $p \leq 0.05$): 450, 462, 406, and 263 sites are available for MAM, JJA, SON, and DJF, respectively, but for each row the relative percentages are shown (i.e., sum to 100%).

| Pct | Season | SNR$\geq$2 $p \leq 0.05$ | 2>SNR$\geq$1 $0.33\leq p <0.05$ | \|SNR\| <1 $p <0.33$ | -2<SNR$\leq$-1 $0.33\leq p <0.05$ | SNR$\leq$-2 $p \leq 0.05$ | Confidence level |
|---|---|---|---|---|---|---|---|
| 90th | MAM | 0.4 | 1.8 | 17.6 | 21.1 | 59.1 | High agreement & robust evidence |
| | JJA | 0.9 | 2.6 | 26.6 | 14.9 | 55.0 | High agreement & robust evidence |
| | SON | 2.3 | 5.6 | 26.3 | 19.2 | 46.6 | Medium agreement & robust evidence |
| | DJF | 24.4 | 17.5 | 42.2 | 9.9 | 6.2 | Medium agreement & limited evidence |
| 50th | MAM | 2.7 | 4.0 | 26.9 | 28.9 | 37.7 | Medium agreement & robust evidence |
| | JJA | 2.1 | 4.1 | 25.5 | 13.6 | 54.5 | High agreement & robust evidence |
| | SON | 14.0 | 14.3 | 41.1 | 20.4 | 10.1 | Medium agreement & limited evidence |
| | DJF | 34.6 | 16.3 | 42.2 | 5.7 | 1.2 | Medium agreement & limited evidence |
| 10th | MAM | 6.5 | 9.1 | 50.9 | 23.3 | 10.2 | High agreement & limited evidence |
| | JJA | 5.5 | 6.4 | 35.4 | 27.0 | 25.5 | Medium agreement & limited evidence |
| | SON | 25.1 | 18.2 | 47.2 | 7.4 | 2.2 | Medium agreement & limited evidence |
| | DJF | 41.4 | 17.5 | 33.8 | 6.5 | 0.4 | Medium agreement & robust evidence |



**Table A3.** Same format as Table A2, but the following before/after scenarios are categorized:

(A) P→N: from reliable positive ($p \leq 0.05$) to reliable negative ($p \leq 0.05$) trends;

(B) W→N: from weak ($p > 0.05$) to reliable negative ($p \leq 0.05$) trends;

(C) N→P: from reliable negative ($p \leq 0.05$) to reliable positive ($p \leq 0.05$) trends;

(D) W→P: from weak ($p > 0.05$) to reliable positive ($p \leq 0.05$) trends;

(E) P→W: from reliable positive ($p \leq 0.05$) to weak ($p > 0.05$) trends;

(F) N→W: from reliable negative ($p \leq 0.05$) to weak ($p > 0.05$) trends;

All the other scenarios (e.g. the same reliability scale before/after the changepoint, or transitions between weak positive and weak negative) are considered to be no evidence of trend changes.

| Pct | Season | P→N | W→N | N→P | W→P | P→W | N→W | No evidence |
|------|--------|------|------|------|------|------|------|-------------|
| 90th | MAM | 3.1 | 53.8 | 0.2 | 0.2 | 2.2 | 6.4 | 34.0 |
| | JJA | 0.4 | 50.3 | 0.2 | 0.7 | 1.1 | 27.7 | 19.6 |
| | SON | 1.5 | 43.8 | 0.5 | 1.5 | 0.5 | 16.5 | 35.7 |
| | DJF | 2.7 | 3.4 | 1.1 | 21.7 | 8.7 | 2.3 | 60.1 |
| 50th | MAM | 12.0 | 25.3 | 0.2 | 2.4 | 9.8 | 3.8 | 46.4 |
| | JJA | 1.1 | 49.8 | 0.6 | 1.5 | 1.5 | 16.0 | 29.4 |
| | SON | 1.5 | 8.4 | 2.2 | 10.6 | 7.6 | 7.6 | 62.1 |
| | DJF | 1.1 | 0 | 1.1 | 29.7 | 21.7 | 0.4 | 46.0 |
| 10th | MAM | 4.4 | 5.8 | 0 | 6.2 | 21.8 | 1.3 | 60.4 |
| | JJA | 1.7 | 23.6 | 0 | 5.6 | 3.2 | 7.1 | 58.8 |
| | SON | 0.5 | 1.7 | 3.7 | 20.9 | 14.0 | 1.7 | 57.5 |
| | DJF | 0.4 | 0.4 | 3.4 | 36.1 | 24.7 | 0.4 | 34.6 |