# Peer review of "Surface ozone trend variability across the United States and the impact of heatwaves (1990-2023)"

_EGUsphere, 2024_

## Community Comment (CC1)

Comments by Rodrigo J. Seguel on behalf of the TOAR-II Steering Committee on:

**Surface ozone trend variability across the United States and the impact of heatwaves (1990-2023)**

Kai-Lan Chang (corresponding author), Brian C. McDonald, Owen R. Cooper

This manuscript was submitted to ACP as part of the TOAR-II Community Special Issue
https://doi.org/10.5194/egusphere-2024-3674
Discussion started: 9 December 2024; discussion closes 20 January 2025

This review is by Rodrigo Seguel, member of the TOAR-II Steering Committee. The primary purpose of these reviews is to identify any discrepancies across the TOAR-II submissions, and to allow the author teams time to address the discrepancies. Additional comments may be included with the reviews.

While members of the TOAR Steering Committee may post open comments on papers submitted to the TOAR-II Community Special Issue, they are not involved with the decision to accept or reject a paper for publication, which is entirely handled by the journal's editorial team.

**General comments**

The authors provide a comprehensive trend assessment of surface ozone across the USA (1990-2023). The effectiveness of interventions, i.e., control emissions, was quantified on ozone trends using a changepoint detection algorithm. Also, the authors investigated the potential impacts of heatwave events on ozone exceedances. The methodology described in the manuscript and the rationale behind the statistical methods applied are clear and sound.

Overall, the results are consistent with other reports that showed decreasing ozone trends across the USA (stronger in the eastern USA since the 2000s), attributed to strict controls of anthropogenic emissions. In addition, the study shows the impact of heat waves on increasing ozone exceedance in California, thus counteracting the effectiveness of emission controls.

Therefore, this paper significantly contributes to the TOAR community's objective of reliably quantifying ozone trends and attribution.

**Minor comment**

In the paper, mixing ratios are reported in units of ppbv. However, in the TOAR phase II, we have adopted the units of nmol mol$^{-1}$ instead of ppbv to express mixing ratios. In some cases, mainly related to human exposure, keeping the units of ppbv has been preferred to maintain consistency between the unit's metrics. Therefore, I suggest including a short explanation or clarification about the decision to use ppbv in this study.

---

## Author Comment (AC1)

We thank the referees for the obvious time and care they put into their reviews, which helped us to revise the manuscript with improved focus and clarity. We have addressed all of the referee comments as described below. The reviewer comments are shown in bold font, followed by our response in normal font.

**Anonymous Referee #1**
**This paper presents results for surface ozone trends in the United States over the period from 1990-2023 and investigates the relationship between heatwaves and ozone over time. Surface ozone trends and their relationship to heatwaves are an important topic in air quality. This study incorporates novel approaches such as changepoint detection and a variety of statistical metrics to provide new insights on a long-standing topic.**

**General Comments**
1. **Either within the results or in the conclusions, it would be useful to expand on how the results of this study fit within the context of the large existing literature on US surface ozone trends. Please highlight what is new versus confirmation of previous findings.**

Thank you for the suggestion. We have added context for US ozone studies, and have rewritten the text in the Conclusions as follows:
*"Previous studies on US surface ozone observations have shown a substantial reduction in warm-season ozone across much of the US since the early 2000s, in response to strict controls of anthropogenic emissions (Cooper et al., 2012; Simon et al., 2015; Strode et al., 2015; Lin et al., 2017; Jin et al., 2020; Seltzer et al., 2020; Simon et al., 2024). However, several outstanding issues are noted, including, (1) oil and gas emissions (McDuffie et al., 2016; Francoeur et al., 2021), which can contribute to wintertime ozone exceedances in the Uinta Basin, UT (Edwards et al., 2014); (2) wildfire influence, which has been associated with regional-scale ozone exceedances in recent years (Langford et al., 2023; Byrne et al., 2024; Cooper et al., 2024); and (3) heatwaves, which provide ideal conditions for ozone production (Schnell and Prather, 2017). By quantifying trends based on all available MDA8 ozone observations over the extended period 1990-2023, we found that our analysis is consistent with prior studies, but we have been able to update trends to the present day, and for the first time we provide observational evidence of the ozone climate penalty in regions with large temperature increases and widespread ozone monitoring."*

We also highlighted our new findings as follows:
*"In the eastern USA, despite that flattened JJA trends are observed after 2013, the added value of our long-term study is to show that the rapidly declining JJA trends in the 2000s substantially reshaped the regional ozone season, with the seasonal peak shifting from summer in the 1990s to spring in the most recent decade."*

All the references above are cited in the initial submission, so we do not list those references here, except for:

Simon, H., Hogrefe, C., Whitehill, A., Foley, K. M., Liljegren, J., Possiel, N., Wells, B., Henderson, B. H., Valin, L. C., Tonnesen, G., Appel, K. W., and Koplitz, S.: Revisiting day-of-week ozone patterns in an era of evolving US air quality, Atmos. Chem. Phys., 24, 1855–1871, https://doi.org/10.5194/acp-24-1855-2024, 2024.

2. **On page 7, it is noted that all changepoint candidates are considered. However, we might have an a priori expectation that the changepoint will occur in a particular year (or range of years) given known changes in emissions. What is the justification for considering all possible years as candidates?**

From a rigorous statistical perspective, we indeed consider a range of years as changepoint candidates, however we only consider changepoints within the period 2000-2013 when emissions inventories show steep reductions in NOx emissions. Specifically,

1. Our consideration is to allow a benchmark period for identifying a change in trend (i.e., at least 10-years of a persistent pattern, before and after the changepoint). This implies that we avoid assigning a changepoint during the early period (1990-1999) and the late period (2014-2023), because it is difficult to discern a change of long-term trends from interannual variability, if the changepoint is found near the beginning or end of the time series. This also implies that we have sufficient statistical evidence to conclude a changepoint of long-term trends, if the changepoint occurs between 2000 and 2013.
2. This range of years (2000-2013) also includes the expected changes in NOx emission trends, i.e., the rapid decline since the early 2000s.
3. On the other hand, in terms of statistical analysis, the time it takes to detect an effect on ozone from regulation measures depends heavily on the sensitivity of ozone to precursor emissions, the magnitude of ozone interannual variability, and the effectiveness of intervention (local emission changes) at the given location. For example, if the impact is persistent, but very weak, it might take several years to detect a small change above the noise of large interannual variability. This statistical property is noted by Box and Tiao (1975) when they investigated the impact of intervention on ozone pollution in Los Angeles.

Therefore, by taking all of the factors into account, this range of years between 2000-2013 enables us to affirm the reliability of the changepoint, and be flexible enough to allow a delay period between intervention and response.

To point out the potential delay period between regulation intervention and ozone response, we added a discussion in Section 2.2:

*"It should be noted that the impact of emission changes may not be immediately apparent for ozone (Box and Tiao, 1975). The time it takes to detect the impact of regulation measures depends heavily on the sensitivity of ozone to precursor emissions, the magnitude of ozone interannual variability, and the effectiveness of the intervention (local emission changes) at the given location. Therefore, a delay period can be expected, particularly if the magnitude of a trend change is weak compared to the interannual variability."*

To avoid the potential confusion that we consider all years to be changepoint candidates, we revised the text with a more clear explanation in the end of section 2.2:

*"Based on the above discussions, to allow a benchmark period for identifying a change in trend (i.e., at least 10-years of a persistent pattern) and to avoid assigning changepoints near the beginning/end of the time series, we fit the trend model to each tentative changepoint candidate (mainly one or possibly two between 2000-2013), and then select the best fitted result from the candidate pool."*

We also revised the text in Section 2.3 as follows:

*"For each station/season/percentile, the trend model is fitted to each tentative changepoint candidate between 2000 and 2013."*

Reference:
Box, G. E., & Tiao, G. C. (1975). Intervention analysis with applications to economic and environmental problems. Journal of the American Statistical Association, 70(349), 70-79.

3. **Since the ozone changes are often related to emission changes in the text, it would be helpful to include a figure showing regional emissions timeseries. Also, could you apply the changepoint detection methodology to the emissions, and if so, would it yield similar results to ozone concentration changepoints?**

Thanks for the suggestion. We took the total anthropogenic NOx and VOC estimates from EPA's national air pollutant emissions trends dataset (1990-2023) (https://www.epa.gov/air-emissions-inventories/air-pollutant-emissions-trends-data). Note that the EPA emissions trend dataset has a disjuncture in NOx emissions between 2001 and 2002 that is from a methodological change in EPA's mobile source emissions model from MOBILE (Mobile Source Emissions Factor) to MOVES (MOtor Vehicle Emission Simulator). By contrast, we also show NOx emissions where we replace mobile source emissions (on-road + off-road) from the Fuel-Based Inventory of Vehicle Emissions (FIVE), which utilizes a consistent methodology over 1990-2022 and does not exhibit such a disjuncture (McDonald et al., 2014, 2018; Harkins et al., 2021). We then added this new analysis to Section 3.2 as follows:

*"Alternatively, EPA's national air pollutant emissions trends dataset can be used to study total anthropogenic NOx and VOC (volatile organic compounds) emissions (US EPA, 2024b). In addition, the Fuel-Based Inventory of Vehicle Emissions (FIVE) provides improved NOx estimates of mobile emissions (McDonald et al., 2014, 2018; Harkins et al., 2021). FIVE can be used to replace the estimates from "highway vehicles" and "off-highway categories" in the EPA dataset, to produce an updated total anthropogenic NOx emissions inventory. Based on the same changepoint analysis (Fig 4), (1) reliably negative VOC trends (p ≤ 0.01) are found between 1990-2013, but became positive after 2013 (p = 0.09); (2) Although the EPA NOx trends show a reduction in the late 1990s, the first changepoint is identified in 2003, due to a spike in 2002, followed by a rapid decline (p ≤ 0.01) over 2003-2013. The decreasing trends weakened after 2013, but still with high confidence (p ≤ 0.01); and (3) similar to the EPA NOx trends, the FIVE NOx decreased from 1990 to 2022, with an acceleration at 2000 and a*

*deceleration at 2010, which occurred three years earlier than the changepoints in the EPA NOx trends."*

Following this analysis, we then discuss the possibility to include two changepoints if the junctures occur separately at around 2000 and after 2010, and found that the 90th and 50th percentiles in JJA in the eastern USA fit this scenario. The ozone time series in other seasons are so variable that the second changepoint is too premature to determine. The FIVE NOx data are provided and updated by Colin Harkins from University of Colorado Boulder and NOAA Chemical Sciences Laboratory (the 2023 estimate is yet to be available), so we have invited him to join this study.

[Figure]

Fig 4. Annual US NOx and VOC trends: Estimates include EPA's national air pollutant emissions trends for NOx (purple) and VOC (orange). In addition, the FIVE NOx (magenta) emissions are based on the EPA estimates, but the mobile emissions from "highway vehicles" and "off-highway categories" were replaced with the estimates from the Fuel-Based Inventory of Vehicle Emissions (FIVE) (Harkins et al. 2021).

References:
Harkins, C., McDonald, B. C., Henze, D. K., & Wiedinmyer, C. (2021). A fuel-based method for updating mobile source emissions during the COVID-19 pandemic. Environmental Research Letters, 16(6), 065018.

McDonald, B. C., McBride, Z. C., Martin, E. W., & Harley, R. A. (2014). High‑resolution mapping of motor vehicle carbon dioxide emissions. Journal of Geophysical Research: Atmospheres, 119(9), 5283-5298.

McDonald, B. C., McKeen, S. A., Cui, Y. Y., Ahmadov, R., Kim, S. W., Frost, G. J., ... & Trainer, M. (2018). Modeling ozone in the Eastern US using a fuel-based mobile source emissions inventory. Environmental Science & Technology, 52(13), 7360-7370.

US EPA: Air Pollutant Emissions Trends Data, https://www.epa.gov/air-emissions-inventories/air-pollutant-emissions-trends-data, accessed: 02-01-2025, 2024b.

**4. Organization: Section 4.1 doesn't seem specific to heatwaves**

We delegated this section to Appendix B and updated the figure number accordingly throughout the paper. We also revised the text in the beginning of Section 4:

*"The distribution and trends in ozone exceedances based on all daily MDA8 ozone values during May-Sep are provided in Appendix B (without distinguishing normal and heatwave conditions). This section first investigates the short-term heatwave impact on ozone, and then compares long-term ozone exceedances between heatwave and normal days."*

**5. Some statements in the paper need clarification, as noted in the specific comments below.**

Thanks for pointing these out, we addressed each comment explicitly below.

**Specific Comments**

**1. Abstract lines 9-10: please clarify: does this mean decrease in exceedances, or in exceedances during heat waves, or something else?**

We revised the text as follows:

*"When the increasing heatwave trends are accounted for, we find evidence that decreases in exceedances during heatwaves have likely halted at 20%-40% of sites, depending on heatwave definitions."*

**2. On page 4, 6 different temperature threshold definitions are listed. While there is value in considering more than one definition, the text might be clearer if this list were shortened or any sensitivity of the results to the choice of definition highlighted. For example, do we expect TX95pct and TX90pct to yield different results?**

Thanks for pointing this out. We indeed have not discussed why these temperature thresholds are considered, and their expected influence. We added a discussion in Section 2.1:

*"While both $T_{max}$ and $T_{min}$ are important indicators to identify potential heatwaves (Perkins and Alexander, 2013), there is a lack of previous studies addressing the sensitivity of MDA8 exceedance trends based on different heatwave definitions. Although we would expect that a higher $T_{max}$ is typically better correlated with ozone extremes (Porter et al., 2015; Wells et al.,*

*2021), a high temperature threshold might prevent us from having sufficient sample sizes for valid statistical analysis. Therefore, (i) TX35deg represents a fixed threshold, but this threshold is considerably too high for mid- and high latitude sites; (ii) TX95pct and TX90pct enable sufficient observations for studying heatwaves, since these thresholds are site-specific; and (iii) TX95pct provides a warmer condition than TX90pct for ozone production, and can be considered to be a trade-off between TX35deg and TX90pct. In summary, albeit with a much smaller number of sites that can quantify the TX35deg condition, our results show a general similarity between different metrics (see Section 4 for more in-depth discussions)."*

References:
Perkins, S. E. and Alexander, L. V.: On the measurement of heat waves, Journal of Climate, 26, 4500–4517, https://doi.org/10.1175/JCLI-D-12-00383.1, 2013.
Porter, W. C., C. L. Heald, D. Cooley, and B. Russell (2015), Investigating the observed sensitivities of air-quality extremes to meteorological drivers via quantile regression, Atmos. Chem. Phys., 15, 10,349–10,366, doi:10.5194/acp-15-10349-2015.
Wells, B., Dolwick, P., Eder, B., Evangelista, M., Foley, K., Mannshardt, E., Misenis, C., and Weishampel, A.: Improved estimation of trends in US ozone concentrations adjusted for interannual variability in meteorological conditions, Atmos. Environ., 248, 118234, https://doi.org/10.1016/j.atmosenv.2021.118234, 2021

**3. Page 8, lines 4-5: define "average marginal effect"**
We added the definition as follows:
*"Note that the term $\theta_1$ can not directly be interpreted as the trend value (a change per time unit) for exceedance probability, so the average marginal effect is used to represent the slope, i.e., the slope can be represented by calculating the average of the derivatives of the inverse logit function over each time unit t (Kleiber and Zeileis, 2008)."*

**4. Page 10 line 29: what does it mean that positive trends are decreasing? Do you mean they switch from positive to negative, or just become less positive?**
We meant to indicate that trends switch from positive to negative. We revised the text:
*"The overall conclusion is that positive trends were observed in MAM, JJA and SON since 1990, but with varying turnaround points since the 2000s, these trends turned around to be strongly and reliably decreasing…"*

**5. Page 11 bottom: The current peak being in April does not, by itself , prove the cycle is moving toward Feb-March, and it isn't apparent from comparing the different epochs in Fig 4 that it's heading that way**
We revised the text as follows (now Fig 5):
*"The current peak in April suggests that the observed seasonal cycle might follow the model prediction by Clifton et al. (2014), but a modeling study is necessary to confirm the detailed process."*

**6. Page 14 line 8-9: What about the negative arrows in Fig. 7?**
We added the number for the negative accumulations as follows (now Fig 6):

*"Both eastern and western USA have around 10% of sites with reliably decreasing ozone accumulations (p ≤ 0.05)."*

**7. Page 15 line 28: is there a physical mechanism, like a weaker response of O3 to T when NOx is lower?**

Yes, we added a new reference as follows:

*"This result is consistent with a recent study which shows that U.S. emission controls reduce the summertime ozone-temperature sensitivity (Li et al., 2024)."*

Reference:

Li, S., Lu, X., and Wang, H.: Anthropogenic emission controls reduce summertime ozone-temperature sensitivity in the United States, EGUsphere [preprint], https://doi.org/10.5194/egusphere-2024-1889, 2024.

**8. Page 16 line 17-18: I suggest rewording this sentence to make it clear you are contrasting the 41% at the 10th percentile with the lower percentages at the higher percentiles**

We revised the text as follows:

*"In contrast, reliably positive DJF trends are observed more commonly at the lower percentiles: 41% at the 10th percentile, versus 35% at the 50th percentile and 24% at the 90th percentile."*

**9. Page 16 line 21: Does "50th/90th percentiles" mean 50th and 90th percentiles?**

Yes, the text is revised.

**10. Page 16 line 25-27: see general comment 3. This is a place where an emissions timeseries plot or changepoint analysis would be helpful to refer to.**

Thanks for the suggestion. A plot is added in new Fig 4, as discussed in General Comment 3 above.

**11. Page 17 lines 4-5: It's stated that heatwaves likely counteract the effectiveness of emission controls, but aren't the emission controls still effective at reducing the negative impact of heat waves?**

We aim to discuss that reliable decreasing exceedance trends are likely to have halted during heatwaves, so "counteract" might be too strong in this case. We revised the text as follows:

*"Therefore, heatwaves not only increase the exceedance probabilities, but also likely slow down the effectiveness of emission controls."*

We also replace "counteract" with "halt" or "slow down" throughout the paper.

**12. Appendix A first sentence: Either split this sentence in two or add a conjunction/connection between "our focus has been on the seasonal extreme percentile" and "this section aims…"**

This is split into two sentences.

**Anonymous Referee #2**
**This manuscript analyzes changes in ozone trends over the past almost 30 years in the contiguous United States. The authors focus on changepoint detection across various percentiles, and they investigate the impact of potential climate penalties that may offset ozone improvements. They find decreases of extreme ozone values since the 2000s during the warm months, while wintertime trends tend to be increasing. Heat waves increase ozone exceedance probabilities, and some sites in California may have been impacted by the ozone climate penalty during this timeframe.**

**The manuscript and analyses are strong, and this is a good addition to the ozone literature which introduces a statistical technique rarely used in atmospheric chemistry to the analysis of sparse ozone data. There are some minor clarifications and discussions that are needed, discussed below.**

We thank the referee for the positive feedback.

**Page 3, Line 23: "summertime period in which the MDA8 value exceeds the thresholds of 70, 60, 50, and 35 ppbv…" Could you please explain your rationale behind using these ozone concentration cutoffs?**

We explained the cutoff values for ozone exceedances in Section 2.1:
*"In addition to a threshold of 70 ppbv for the US EPA NAAQS, we also report the threshold exceedances based on 60 and 50 ppbv (commonly used to represent non-attainment occurrences for some air quality standards adopted globally, WHO (2006); European Commission (2015); US Federal Register (2015)), as well as 35 ppbv (recommended by the World Health Organization to assess ozone health impacts, WHO (2013))."*

References:
European Commission (2015) Directive 2008/50/EC of the European Parliament and of the Council of 21 May 2008 on ambient air quality and cleaner air for Europe. Available at: http://data.europa.eu/eli/dir/2008/50/2015-09-18.
US Federal Register (2015) National Ambient Air Quality Standards for Ozone. Available at: https://www.federalregister.gov/documents/2015/10/26/2015-26594/national-ambient-air-quality-standards-for-ozone.
WHO (2006) Air Quality Guidelines: Global Update 2005. Particulate matter, ozone, nitrogen dioxide and sulfur dioxide. Available at: https://iris.who.int/bitstream/handle/10665/107823/9789289021920-eng.pdf?sequence=1 .
WHO (2013) Review of evidence on health aspects of air pollution – REVIHAAP Project: technical report. World Health Organization. Regional Office for Europe. Available at: https://iris.who.int/handle/10665/341712.

**Page 4, Line 21: "An episode of heatwave event is detected if at least 3 consecutive days of temperature exceedances are found." What is meant by a temperature exceedance here? Is this peak temperatures, average temperatures, etc.?**

We clarified the temperature exceedances are determined by various thresholds for daily maximum and minimum temperatures as follows:
*"For each temperature threshold described above, a heatwave event is detected if at least 3 consecutive days of corresponding exceedances are found. So a total of six heatwave metrics are considered based on long-term temperature data."*

**Page 5, Line 25 and Page 6, Line 2: "However, the possibility of incorporating two changepoints will be evaluated, if the junctures occur separately at around 2000 and 2010." And "Based on the above discussions, for each monitoring site and season, we fit the trend model to all possible changepoint candidates (between 2000-2013)…" What is the rationale for choosing these years? Is it not possible that changepoints beyond these years could be present?**

It is certainly possible to include changepoints at any given year. However, it is also critical to make sure that we have sufficient evidence to conclude a meaningful trend change. For example, if a changepoint is found with $p < 0.01$ at the end of a time series (e.g. 2021), we shall not be able to distinguish between a truly long-term trend change and merely a short-term variability, because the posterior-trend period (2021-2023) is too short. Further demonstrations can be found in Figs 1 and 2 from Cooper et al. (2020).

Therefore, the rationale is to allow a benchmark period for identifying a change in trend (i.e., at least 10-years of a persistent pattern, before and after the changepoint), which implies that we also need to avoid assigning changepoints near the beginning/end of the time series. The rationale can include two scenarios in our study:
- One changepoint in 1990-2023: the juncture should not occur between 1990-1999 or 2014-2023, and can occur any year between 2000-2013, because this ensures a benchmark of 10-years for both prior-trend and posterior-trend periods.
- Two changepoints in 1990-2023: if the first juncture occurs around 2000, then we can examine the possibility of a second changepoint around 2010, since each trend period satisfies a benchmark of 10-years (1990-~2000, ~2000-~2010, ~2010-2023).

One-changepoint scenario is found appropriate for most cases, and two-changepoints scenario can be applied to the JJA 90th and 50th percentiles in the eastern USA.

We revised the text with a more clear explanation in the end of section 2.2:
*"Based on the above discussions, to allow a benchmark period for identifying a change in trend (i.e., at least 10-years of a persistent pattern) and to avoid assigning changepoints near the beginning/end of the time series, we fit the trend model to each tentative changepoint candidate (mainly one or possibly two between 2000-2013), and then select the best fitted result from the candidate pool."*

Reference:

Cooper, O. R., Schultz, M. G., Schröder, S., Chang, K.-L., Gaudel, A., Benítez, G. C., ... & Xu, X. (2020). Multi-decadal surface ozone trends at globally distributed remote locations. Elem Sci Anth, 8, 23. https://doi.org/10.1525/elementa.420.

**Page 10, Line 6: "This pattern can be attributed to strong ozone enhancements across the Northeast/Midwest in 2012." Does the flattening of the trend in the most recent decade (2013-2023) remain flat if that anomalously high ozone values in 2012 are removed? I am concerned that this one spike early on in the recent time series is drowning out any other trend that may be occurring.**

Thanks for pointing this out. Based on the time series plot in the bottom panel of Fig 1 in the paper, we can expect the JJA trends in the eastern USA to be flat after 2013, which are notably different from (or not related to) the spike in 2012. Nevertheless, we repeated the same analysis by removing the 2012 data in the Northeast (north of 40°N and east of 100°W), and the results showed a great similarity with or without 2012 data (Fig R1). This comparison unexpectedly indicates that the spike in 2012 plays a rather minor role in the quantification of a trend change in the recent decade.

We thus removed the role of strong ozone enhancements across the Northeast/Midwest in 2012, and revised the text as follows:
*"This flattened pattern might be related to the deceleration of NOx emissions reductions since 2010 (Jiang et al., 2018, 2022)."*

The role of NOx trends will be addressed in the next comment.

[Figure]

Fig R1. MDA8 trends prior (first column) and posterior (second column) to the changepoints (third column) for the Jun-Jul-Aug 90th percentile over 1990-2023. The upper panel are the results based on all available data. The same results are shown in the lower panel, but all the 2012 data in the Northeast are removed. Note that the changepoint can occur in 2012 even if the 2012 data are removed, since the changepoint is determined by the overall fitted quality based on all available observations over 1990-2023.

**NOx Trends**
**A flattening of NOx trends after 2010 is mentioned in a few places (Page 10, Line 7-9; Page 11, Line 11-12). It is important to note that the Jiang papers mentioned in the manuscript are based on satellite data, which is sensitive to free tropospheric NOx and may not be fully representative of surface trends. A couple papers come to mind that suggest that NOx trends have not flattened out over urban areas (Christiansen et al., 2024; Silvern et al., 2019). The slowdown in NOx trends after 2010 is most noted over rural areas, where surface NOx is lower and thus background sources of NOx (soil NOx, free tropospheric NOx, lightning NOx) that remain relatively constant over time dominate the signal. Over urban areas, other evidence suggests that NOx emissions have continued to decrease. Since most AQS sites are in urban areas, I am not sure that the ozone trends after 2010 can be adequately explained by a deceleration in NOx decreases.**

We thank the referee for pointing out these studies. We have added these references and revised the text in Section 3.2:
*"However, previous satellite observations showed that US NOx emissions did not decline until the late 1990s (McDonald et al., 2012), and the pace has likely decelerated since 2010 (Jiang et al., 2018, 2022), primarily found in rural areas (Silvern et al., 2019; Christiansen et al., 2024)."*

and in Conclusions:

*"In response to the shift of satellite NOx emissions trends from a rapid decline since the late 1990s to a slowdown after 2010 (McDonald et al., 2012; Jiang et al., 2018, 2022), we explored the possibility of incorporating two changepoints for trend detection. [...] Recent studies have also shown that the slowdown in NOx trends is mainly found over rural areas, while NOx has continued to decrease in urban areas (Silvern et al., 2019; Christiansen et al., 2024). The impact of the most recent NOx trends might take additional years to become apparent and attributable to ozone variability, and warrants future modeling studies."*

**References**
**Christiansen, A., Mickley, L. J., and Hu, L.: Constraining long-term NO$x$ emissions over the United States and Europe using nitrate wet deposition monitoring networks, Atmos. Chem. Phys., 24, 4569–4589, https://doi.org/10.5194/acp-24-4569-2024, 2024.**

**Silvern, R. F., Jacob, D. J., Mickley, L. J., Sulprizio, M. P., Travis, K. R., Marais, E. A., Cohen, R. C., Laughner, J. L., Choi, S., Joiner, J., and Lamsal, L. N.: Using satellite observations of tropospheric NO2 columns to infer long-term trends in US NO$x$ emissions: the importance of accounting for the free tropospheric NO2 background, Atmos. Chem. Phys., 19, 8863–8878, https://doi.org/10.5194/acp-19-8863-2019, 2019.**

**Comments by Rodrigo J. Seguel on behalf of the TOAR-II Steering Committee on:**
**Surface ozone trend variability across the United States and the impact of heatwaves (1990-2023)**
**Kai-Lan Chang (corresponding author), Brian C. McDonald, Owen R. Cooper**

**This manuscript was submitted to ACP as part of the TOAR-II Community Special Issue**
**https://doi.org/10.5194/egusphere-2024-3674**
**Discussion started: 9 December 2024; discussion closes 20 January 2025**
**This review is by Rodrigo Seguel, member of the TOAR-II Steering Committee. The primary purpose of these reviews is to identify any discrepancies across the TOAR-II submissions, and to allow the author teams time to address the discrepancies. Additional comments may be included with the reviews.**
**While members of the TOAR Steering Committee may post open comments on papers submitted to the TOAR-II Community Special Issue, they are not involved with the decision to accept or reject a paper for publication, which is entirely handled by the journal's editorial team.**

**General comments**
**The authors provide a comprehensive trend assessment of surface ozone across the USA (1990-2023). The effectiveness of interventions, i.e., control emissions, was quantified on ozone  trends using a changepoint detection algorithm. Also, the authors investigated the potential impacts of heatwave events on ozone exceedances. The methodology described in the manuscript and the rationale behind the statistical methods applied are clear and sound.**
**Overall, the results are consistent with other reports that showed decreasing ozone trends across the USA (stronger in the eastern USA since the 2000s), attributed to strict controls of anthropogenic emissions. In addition, the study shows the impact of heat waves on increasing ozone exceedance in California, thus counteracting the effectiveness of emission controls.**
**Therefore, this paper significantly contributes to the TOAR community's objective of reliably quantifying ozone trends and attribution.**

**Minor comment**
**In the paper, mixing ratios are reported in units of ppbv. However, in the TOAR phase II, we have adopted the units of nmol mol$^{-1}$ instead of ppbv to express mixing ratios. In some cases, mainly related to human exposure, keeping the units of ppbv has been preferred to maintain consistency between the unit's metrics. Therefore, I suggest including a short explanation or clarification about the decision to use ppbv in this study.**

Thanks for the positive feedback. We added a short clarification about the units in Section 2.1:
*"It should be noted that TOAR uses the mole fraction of ozone in air (nmol mol$^{-1}$) to describe the mixing ratio of an ozone observation. Whereas in order to maintain consistency with the human health research community, the units of parts per billion by volume (ppbv) are used to report MDA8, as recommended by the TOAR guidelines."*